# The use of personal weather station observations for improving precipitation estimation and interpolation

András Bárdossy[1], Jochen Seidel[1], and Abbas El Hachem[1]

[1]Institute for Modelling Hydraulic and Environmental Systems, University of Stuttgart, D-70569 Stuttgart, Germany

**Correspondence:** Jochen Seidel (jochen.seidel@iws.uni-stuttgart.de)

**Abstract.** The number of personal weather stations (PWS) with data available online through the internet is increasing gradually in many parts of the world. The purpose of this study is to investigate the applicability of these data for the spatial interpolation of precipitation using a novel approach based on indicator correlations and rank statistics. Due to unknown errors and biases of the observations rainfall amounts from the PWS network are not considered directly. Instead, it is assumed that the temporal order of the ranks of these data is correct. The crucial step is to find the stations which fulfil this condition. This is done in two steps, first by selecting the locations using time series of indicators of high precipitation amounts. The remaining stations are then checked whether they fit into the spatial pattern of the other stations. Thus, it is assumed that the quantiles of the empirical distribution functions are accurate.

These quantiles are then transformed to precipitation amounts by a quantile mapping using the distribution functions which were interpolated from the information from German National Weather Service (DWD) data only. The suggested procedure was tested for the State of Baden-Württemberg in Germany. A detailed cross validation of the interpolation was carried out for aggregated precipitation amounts of 1, 3, 6, 12 and 24 hours. For each of these temporal aggregations, nearly 200 intense events were evaluated and the improvement of the interpolation was quantified. The results show that filtering of observations from PWS is necessary as the interpolation error after filtering and data transformation decreases significantly. The biggest improvement is achieved for the shortest temporal aggregations.

## 1 Introduction

Comprehensive reviews on the current state of citizen science in the field of hydrology and atmospheric sciences were published by Buytaert et al. (2014) and Muller et al. (2015). Both of these reviews give a detailed overview of the different forms of citizen science data and highlight the potential to improve knowledge and data in the fields of hydrology and hydro-climatology. One type of information which is of particular interest for hydrology are data from in-situ sensors. In recent years, the number of low-cost personal weather stations (PWS) has increased considerably. Data from PWS are published online on internet portals such as Netatmo (www.netatmo.com) or Weather Underground (www.wunderground.com). These stations provide weather observations which are available in real time as well as for the past. This is potentially very useful to complement systematic weather observations of national weather services, especially with respect to precipitation, which is highly variable in space and time.

Traditionally rainfall is interpolated using point observations. The shorter the temporal aggregation the higher the variability of rainfall becomes, and the more the quality of interpolation deteriorates (Bárdossy and Pegram, 2013; Berndt and Haberlandt, 2018). In consequence, the number of interpolated precipitation products with sub-daily resolution is low, but such data are required for many hydrological applications (Lewis et al., 2018). Additional information such as radar measurements can improve interpolation (Haberlandt, 2007), however, radar rainfall estimates are still highly prone to different kinds of errors (Villarini and Krajewski, 2010) and the time periods where radar data is available are still rather short.

Against the backdrop of low precipitation station densities, the additional data from PWS has a high potential to improve the information of spatial and temporal precipitation characteristics. However, one of the major drawbacks from PWS precipitation data is their trustworthiness. There is little systematic control on the placing and correct installation and maintenance of the PWS, so it is usually not known whether a PWS is set up according to the international standards published by the WMO (World Meteorological Organization, 2008). Furthermore, there's no information available about the maintenance of PWS. Therefore, precipitation data from PWS may contain numerous errors resulting from incorrect installation, poor maintenance, faulty calibration and data transfer errors (de Vos et al., 2017). This shows that the data from PWS networks cannot be regarded to be as reliable as those of professional networks operated by national weather services or environmental agencies. Consequently, the use of PWS data requires specific efforts to to detect and take these errors into account.

For air temperature measurements, Napoly et al. (2018) developed a quality control (QC) procedure to filter out suspicious measurements from PWS stations that are caused e.g. by solar exposition or incorrect placement. For precipitation, de Vos et al. (2017) investigated the applicability of personal stations for urban hydrology in Amsterdam, Netherlands. They reported results of a systematic comparison of an official observation of the Royal Netherlands Meteorological Institute (KNMI) and three PWS Netatmo rain gauges. This provides information on the quality of measurements in case of correct installation of the devices. As many of the PWS may be placed without consideration of the WMO standards, the results of these comparisons cannot be transferred to the other PWS observations. In a more recent study, de Vos et al. (2019) developed a QC methodology of PWS precipitation measurements based on filters which detect faulty zeroes, high influxes and stations outliers based on a comparison between neighbouring stations. A subsequent bias correction is based on a comparison of past observations with a combined rain gauge and radar product (de Vos et al., 2019).

Overall, the data from PWS rain gauges may provide useful information for many precipitation events and may also be useful for real-time flood forecasting, but data quality issues have to be overcome. In this paper we focus on the use of PWS data for the interpolation of intense precipitation events. We propose a two-fold approach based on indicator correlations and spatial patterns to filter out suspicious measurements and to use the information from PWS indirectly. The basic assumption hereby is that many of the stations may be biased but are correct in the temporal order. For the spatial pattern, information from a reliable precipitation network, e.g. from a national weather service is required. These measurements are considered to be more trustworthy than the PWS data, however, the number of such stations is usually much lower. This paper is organized as follows: After the introduction, the methodology to find useful information and the subsequent interpolation steps are described. The described procedure was used for precipitation events of the last four years in the federal state of Baden-Württemberg in South-

60 West Germany. The results of the interpolation and the corresponding quality of the method are discussed in section 4. The paper ends with a discussion and conclusions.

## 2 Study Area and Data

The federal state of Baden-Württemberg is located in South-West Germany and has an area of approximately 36,000 km$^2$. The annual precipitation varies between 600 and 2,100 mm (Deutscher Wetterdienst, 2020), and the highest amounts are recorded

in the higher elevations of the mountain ranges of the Black Forest. The rain gauge network of the German Weather Service (DWD) in Baden-Württemberg (referred to as primary network from here on) currently comprises 111 stations for the study period with high temporal resolution data (Fig. 1). The gauges used in this network are predominantly weighing gauges. This precipitation data is available in different temporal resolutions from the Climate Data Center of the DWD. For this study, hourly precipitation data was used.

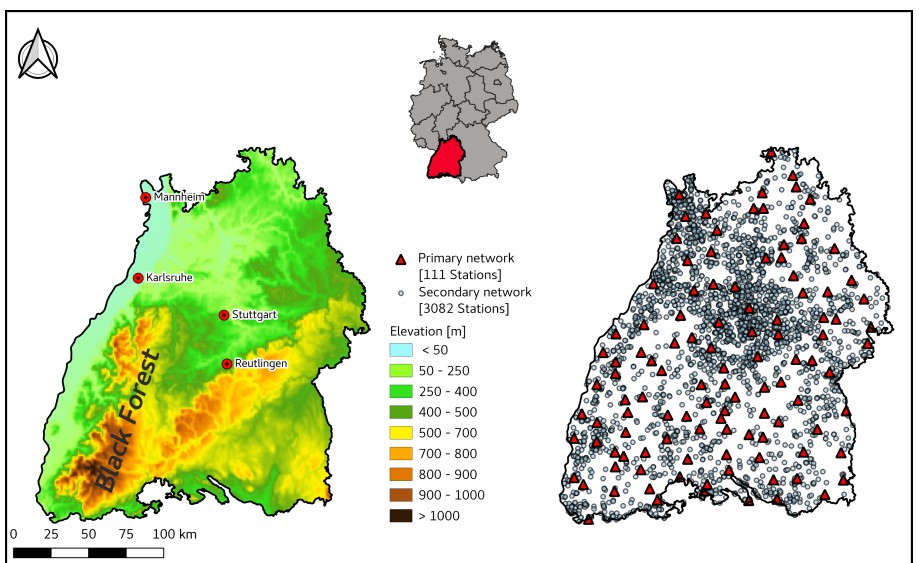

**Figure 1.** Map of the federal state of Baden-Württemberg showing the topography and the location of the DWD (primary) and Netatmo (secondary) gauges.

For the PWS data, the Netatmo network was selected (https://weathermap.netatmo.com). The stations from this PWS network (referred to as secondary network from here onwards) show an uneven distribution in space, which mainly reflects the population density and topography of the study area (Fig. 1). The number of secondary stations is higher in densely populated areas such as in the Stuttgart metropolitan area and the Rhine-Neckar Metropolitan Region between Karlsruhe and Mannheim. Furthermore, there are no secondary network stations above 1,000 m a.s.l., however the primary network only has one station

above 1,000 m (at the Feldberg summit at 1,496 m) as well. The number of gauges from the secondary network varies over time. The time period from 2015 to 2019 was considered for this study, as before 2015 the number of available PWS was very

low. At the end of this time period over 3,000 stations from the secondary network were available. Figure 2 shows the number of secondary stations as a function of time and the length of the time series. One can see that many stations have less than one year of observations, which is the reasonable length of a series for the suggested method. Presently it cannot accommodate series shorter than a year (excluding time periods with snowfall), but as the series are getting longer more and more PWS observations become useful.

The Netatmo rain gauges are plastic tipping buckets which have an opening orifice of 125 cm$^2$ (compared to 200 cm$^2$ of the primary network). A detailed technical description of the Netatmo PWS is given by de Vos et al. (2019). Since these devices are not heated, their usage is limited to liquid precipitation. To take this into account, data from secondary stations were only used in case the average daily air temperature at the nearest DWD station was above 5 °C. Data from the Netatmo PWS network can be downloaded with the Netatmo API either as raw data with irregular time intervals or in different temporal resolutions down to 5 minutes. Further information on how the raw data are processed to different temporal aggregations is not available on the manufacturer's website. For this study, the hourly precipitation data from the Netatmo API was used.

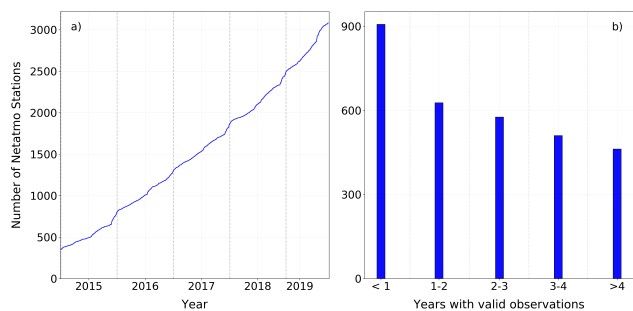

**Figure 2.** Development of the number of online available Netatmo rain gauges (a) and length of available valid hourly observations in Baden-Württemberg (b).

In order to assess the spatial variability within a dense network of primary gauges, the precipitation data from the municipality of Reutlingen (located about 30 km south of the state capital Stuttgart) was additionally used. This city operates a dense network of 12 weighing rain gauges (OTT Pluvio$^2$) since 2014 in an area of 87 km$^2$ (not shown in Fig 1). Furthermore, three Netatmo rain gauges were installed at the Institute's own weather station on the Campus of the University of Stuttgart, where a Pluvio$^2$ weighing rain gauge is installed as well. This allows a direct comparison between the gauges from the primary network and the secondary network in the case the latter are installed and maintained correctly.

## 3  Methodology

It is assumed that the secondary stations may have individual measurement problems, (e.g. incorrect placement, lack of and/or wrong maintenance, data transmission problems) and due to their large number there is no possibility to check their proper placing and functioning directly. Furthermore, at many locations (especially in urban areas) there is no possibility to set up the

rain gauges in such a way that they fulfil the WMO standards. Therefore, the goal is to filter out stations which deliver data
contradicting the observations of the primary network which meet the WMO standards.

Observations from the primary and secondary network were used in hourly time steps and can be aggregated to different durations $\Delta t$. The usefulness of the secondary data is investigated for different temporal aggregations. $Z_{\Delta t}(x,t)$ is the (partly unknown) precipitation at location $x$ and time $t$ integrated over the time interval $\Delta t$. It is assumed that this precipitation is measured by the primary network at locations $\{x_1, \ldots, x_N\}$. The measurements of the secondary network are indicated
as $Y_{\Delta t}(y_j, t)$ at locations $\{y_1, \ldots, y_M\}$. Note that $Y$ is not considered to be a spatially stationary random field. The basic assumption for the suggested quality control and bias correction method is that the measured precipitation data from the secondary network may be biased in their values but correct in terms of their order - at least for high precipitation intensities. This means that if at times $t_1$ and $t_2$:

$$Y_{\Delta t}(y_i, t_1) < Y_{\Delta t}(y_i, t_2) \Rightarrow Z_{\Delta t}(y_i, t_1) < Z_{\Delta t}(y_i, t_2) \tag{1}$$

This means that the measured precipitation amount from the secondary network is likely to have an unknown location specific bias, but the order of values at a location is preserved. This assumption is reasonable specifically for high precipitation intensities and supported by measurements presented in the results section.

For QC two filters are applied. The first one is an indicator based filter (IBF) which compares the secondary time series with the closest primary series with the focus on intense precipitation. The precipitation values of the remaining PWS stations
are then bias corrected using quantile mapping. The second filter is an event based filter (EBF) designed to remove individual contradicting observations for a given time step using a spatial comparison. These two filters and the bias correction are described in the following sections.

### 3.1 High intensity indicator based filtering (IBF)

As a first step in quality control, all PWS with notoriously inconsistent rainfall values are removed. For this purpose the
dependence between neighbouring stations is investigated.

In order to identify stations which are likely to deliver reasonable data for high intensities, indicator correlations are used. The distribution function of precipitation at location $x$ is denoted as $F_{x,\Delta t}(z)$ and the one for secondary observations at locations $y_j$ as $G_{y_j,\Delta t}(z)$, respectively. For a selected probability $\alpha$ the indicator series

$$I_{\alpha,\Delta t,Z}(x,t) = \begin{cases} 1 & \text{if } F_{x,\Delta t}\left(U_{\Delta t}(x,t)\right) > \alpha \\ 0 & \text{else} \end{cases} \tag{2}$$

and for a secondary location $y_j$

$$I_{\alpha,\Delta t,Y}(y_j,t) = \begin{cases} 1 & \text{if } G_{y_j,\Delta t}\left(Y_{\Delta t}(y_j,t)\right) > \alpha \\ 0 & \text{else} \end{cases} \tag{3}$$

Under the order assumptions of equation (1), for any secondary location $y_j$ the two indicator series are identical $I_{\alpha,\Delta t,Z}(y_j,t) = I_{\alpha,\Delta t,Y}(y_j,t)$. Thus the spatial variability of $I_{\alpha,\Delta t,Z}$ and $I_{\alpha,\Delta t,Y}$ has to be the same.

For any two locations corresponding to the primary network $x_i$ and $x_j$ and any $\alpha$ and $\Delta t$ the correlation (in time) of the indicator series is $\rho_{Z,\alpha,\Delta t}(x_i, x_j)$ and provides an information on how precipitation series vary in space. This indicator correlation usually decreases with increasing separation distance. This decrease is not at the same rate everywhere and not the same for different thresholds and aggregations. For the secondary network, indicator correlations $\rho_{Z,Y,\alpha,\Delta t}(x_i, y_j)$ with the series in the primary network can be calculated. Following the hypothesis from equation (1), these correlations should be similar and can be compared to the indicator correlations calculated from pairs of the primary network.

The sample size has a big influence on the variance of the indicator correlations. Therefore, to take into account the limited interval of availability of the secondary observations, indicator correlations of the primary network corresponding to the same periods for which the secondary variable is available are used for the comparison. This is done individually for each secondary site. A secondary station is flagged as suspicious if its indicator correlations with the nearest primary network points are below the lowest indicator correlation corresponding to the primary network for the same time steps and at the nearly same separation distance. A certain tolerance $\Delta_d$ for the selection of the pairs of the primary network is needed due to the irregular spacing of the secondary stations and the natural variability of precipitation. This means if:

$$\rho_{Z,Y,\alpha,\Delta t}(x_i, y_j) < \min\{\rho_{Z,\alpha,\Delta t}(x_k, x_m)\, ; \, \|(x_k - x_m) - (x_i - y_j)\| < \Delta_d\} \tag{4}$$

then the secondary station shows weaker association to the primary than what one would expect from primary observations. In this case it is reasonable to discard the measured time series corresponding to the secondary network at location $y_i$. This procedure can be repeated for a set of selected $\alpha$ values.

Under the assumption that the temporal order of precipitation at secondary locations is correct (eq. 1), one could have used rank correlations instead of the indicator correlations. The indicator approach is preferred however, as the sensitivity of the devices of the primary and secondary networks is different and this would influence the order of the small values strongly. Furthermore, random measurement errors would also influence the order of low values. In order to have a sufficient sample size and to have robust results, high $\alpha$ values and low temporal aggregations $\Delta t$ are preferred.

## 3.2 Bias correction: Precipitation amount estimation for secondary observations

After the selection of the potentially useful secondary stations the next step is to correct their observations. The assumption in equation (1) means that the measured precipitation amounts from the secondary network are likely to have an unknown bias, but the order of values at a location is preserved. This assumption is likely to be reasonable for high precipitation intensities. Thus, the percentile of the precipitation observed at a given time at a secondary location can be used for the estimation of the *true* precipitation amounts. Since this is a percentile and not a precipitation amount it has to be converted to a precipitation amount for further use. This can be done using the distribution function of precipitation amounts corresponding to the location $y_j$ and the aggregation $\Delta t$. As the observations from the secondary network could be biased their distribution $G_{y_j,\Delta t}$ cannot be used for this purpose. Thus, one needs an unbiased estimation of the local distribution functions.

Distribution functions based on long observation series are available for the locations of the primary network. For locations of the secondary network they have to be estimated via interpolation. This can be done by using different geostatistical methods.

A method for interpolating distribution functions for short aggregation times is presented in Mosthaf and Bardossy (2017). Another possibility is to interpolate the quantiles corresponding to selected percentiles or interpolating percentiles for selected precipitation amounts. Another option to estimate distribution functions corresponding to arbitrary locations is to use functional Kriging (Giraldo et al., 2011) to interpolate the distribution functions directly. The advantage of interpolating distribution functions is that they are strongly related to geographical locations of the selected location and to topography. These variables are available in high spatial resolution for the whole investigation domain. Additionally, observations from different time periods and temporal aggregations can also be taken into account as co-variates.

In this paper Ordinary Kriging (OK) is used for the interpolation of the quantiles and for the percentiles to construct the distribution functions both for the locations of the secondary observations and for the whole interpolation grid. For a given temporal aggregation $\Delta t$, time $t$ and target secondary location $y_j$ the observed percentile of precipitation is:

$$P_{\Delta t}(y_j, t) = G_{y_j, \Delta t} \left( Y_{\Delta t}(y_j, t) \right) \tag{5}$$

For the observations of the primary network the quantiles of the precipitation distribution at the primary stations are selected. The distributions at the primary stations are based on the same time steps as those which have valid observations at the target secondary station. In this way, a possible bias due to the short observation period at the secondary location can be avoided. The quantiles are:

$$Q_{\Delta t}(x_i) = F_{\Delta t, x_i}^{-1} \left( P_{\Delta t}(y_j, t) \right) \tag{6}$$

These quantiles are interpolated using OK to obtain an estimate of the precipitation at the target location.

$$Z_{\Delta t}^{o}(y_j, t) = \sum_{i=1}^{n} \lambda_i Q_{\Delta t}(x_i) \tag{7}$$

Here the $\lambda_i$-s are the weights calculated using the Kriging equations. Note that the precipitation amount at the target location is obtained via interpolation, but the interpolation is not using the primary observations corresponding to the same time, but instead is using the quantiles corresponding to the percentile of the target secondary station observation. Thus, these values may exceed all values observed at the primary stations at time $t$. Note that this correction of the secondary observations is non-linear. This procedure is used for all locations which were accepted after application of the indicator filter. In this way, the bias from observed precipitation values at the secondary stations is removed using the observed percentiles and the distributions at the primary stations. This transformation does not require an independent ground truth of best estimation of precipitation at the secondary locations.

## 3.3 Event based spatial filtering (EBF)

While some stations may work properly in general, due to unforeseen events (such as battery failure or transmission errors) they may deliver individual faulty values at certain times. In order to filter out these errors a simple geostatistical outlier detection method is used as described in Bárdossy and Kundzewicz (1990). The geostatistical methods used for outlier detection and

the interpolation of rainfall amounts require the knowledge of the corresponding variogram. However, the highly skewed distribution of the precipitation amounts makes the estimation of the variogram difficult. Instead one can use rank based methods for this purpose as suggested in Lebrenz and Bárdossy (2017) and rescale the rank based variogram.

For a given temporal aggregation $\Delta t$, time $t$ and target secondary location $y_j$ the precipitation amount is estimated via OK using the observations of aggregation $\Delta t$ at time $t$ of primary stations. This value is denoted as $Z^*_{\Delta t}(y_j, t)$. If the precipitation amount at the secondary station estimated using equation (7) differs very much from $Z^*_{\Delta t}(y_j, t)$, the secondary location is discarded for the interpolation. As limit for the difference, three times the Kriging standard deviation was selected. Formally:

$$\left| \frac{Z^*_{\Delta t}(y_j, t) - Z^o_{\Delta t}(y_j, t)}{\sigma_{\Delta t}(y_j, t)} \right| > 3 \tag{8}$$

This means that if the estimated precipitation at the secondary location does not fit into the pattern of the primary observations then it is discarded. Note that this filter is not necessarily discarding secondary observations which differ from the primary - it only removes those where there is a strong local disagreement. This procedure is predominantly removing false zeros at secondary observations which are e.g. due to temporary loss of connection between the rain gauge module and the Netatmo base station.

## 3.4   Interpolation of precipitation amounts

After the application of the two filters and the bias correction the remaining PWS data can be used for spatial interpolation. Once the percentiles of the secondary locations are converted to precipitation amounts, different Kriging procedures can be used for the interpolation over a grid in the target region. The simplest solution is to use OK. For aggregations of one day or longer, the orographic influence should be taken into account. This can be done by using External Drift Kriging (Ahmed and de Marsily, 1987).

A problem that remains when using these Kriging procedures is that the precipitation amounts of the secondary network are more uncertain than those of the primary network. To reflect this difference, a modified version of Kriging as described in Delhomme (1978) is applied. This allows for a reduction of the weights for the secondary stations.

Suppose that for each point $y_i$ time $t$ and temporal aggregation $\Delta t$ there is an unknown error of the percentiles $\varepsilon(y_i, t)$ which has the following properties:

1. Unbiased :

$$E[\varepsilon(y_i, t)] = 0 \tag{9}$$

2. Uncorrelated :

$$E[\varepsilon(y_i, t)\varepsilon(y_j, t)] = 0 \text{ if } i \neq j \tag{10}$$

3. Uncorrelated with the parameter value:

$$E[\varepsilon(y_i, t)Z(y_i, t)] = 0 \tag{11}$$

For the primary network we assume that $\varepsilon(x_i, t) = 0$.

The interpolation is based on the observations

$$\{u_1, \ldots, u_N\} = \{x_1, \ldots, x_N\} \cup \{y_1, \ldots, y_M\} \tag{12}$$

For any location $x$

$$Z_{\Delta t}^*(x, t) = \sum_{i=1}^{n} \lambda_i \left( Z(u_i, t) + \varepsilon(u_i, t) \right) \tag{13}$$

To minimize the estimation variance an equation system similar to the OK system has to be solved, namely:

$$\sum_{j=1}^{n} \lambda_j \gamma(u_i - u_j) + \lambda_i E[\varepsilon(u_i, t)^2] + \mu = \gamma(u_i - x) \ \ i = 1, \ldots, n$$

$$\sum_{j=1}^{n} \lambda_j = 1 \tag{14}$$

Note that OK is a special case of this procedure with the additional assumption $\varepsilon(y_j, t) = 0$. This system leads to an increase of the weights for the primary and a decrease of the weights for the secondary network. For each time step and percentile the variances of the random error terms $\varepsilon(y_i, t)$ is estimated from the interpolation error of the distribution functions. This interpolation method is referred to as Kriging using uncertain data (KU) (Delhomme, 1978). The variograms used for interpolation were calculated in the rank space using the observations of the primary network only which leads to more robust results.

(Lebrenz and Bárdossy, 2017). Anisotropy was not considered, the main reason for this was that the primary network did not give robust results.

## 3.5   Step by step summary of the methodology

In summary, the procedure for using secondary observations is as follows:

1. Select a percentile threshold for a selected temporal aggregation. The threshold should be adapted to the temporal aggregation, e.g. 98 or 99 % for hourly or 95 % for 3 hourly data.

2. Calculate the indicator series for primary and secondary stations corresponding to the percentile threshold.

3. For each individual secondary station:

   (a) Calculate the indicator correlation of the given secondary and the closest primary station.

   (b) Calculate the indicator correlations of all primary stations using data corresponding to the time steps of the selected secondary station.

   (c) Compare the correlations and keep the secondary station if its indicator correlation is in the same range as the indicator correlations of the primary stations approximately at the same distance (IBF).

4. Perform a bias correction by interpolating the distribution function values of the primary network.

5. Select an event to be interpolated and calculate the corresponding variogram of precipitation (based on rank statistics).

    (a) Calculate the percentile of observed precipitation (based on the corresponding time series).

    (b) Calculate the quantiles corresponding to the above secondary percentile for the closest $M$ primary stations of observed precipitation (based on the corresponding time series).

    (c) Interpolate the quantiles for the location of the secondary station using the above primary values using OK, and assign the obtained value to the secondary location.

6. Interpolate precipitation for each secondary location using OK excluding the value assigned to the location (cross validation mode).

7. Compare the interpolated and the assigned (5.c) value and remove station if condition of inequality (eq. 8) indicates outlier.

8. Interpolate precipitation for target grid using all remaining values .

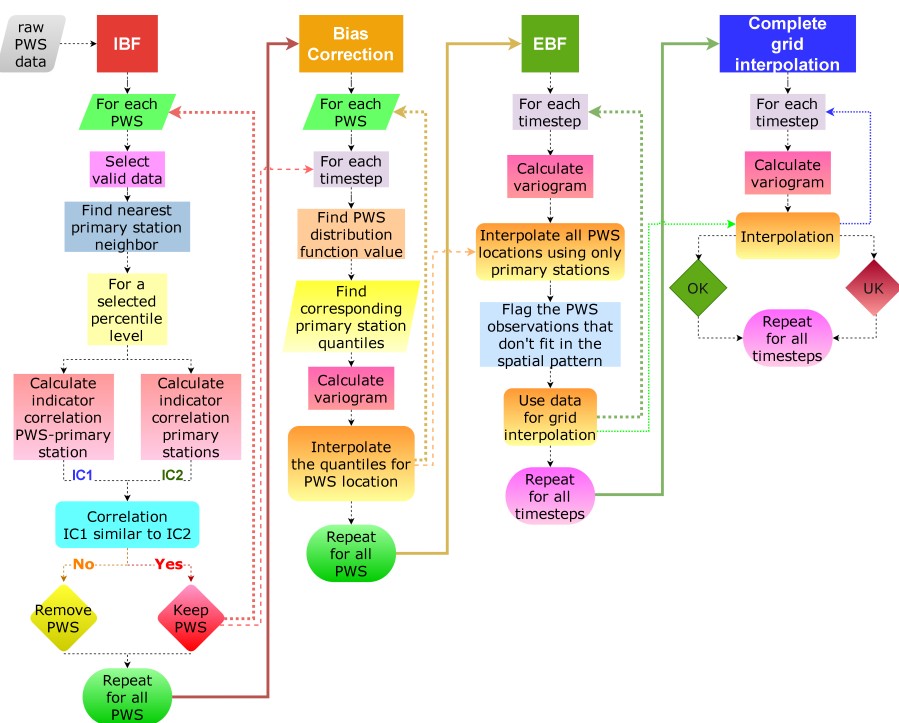

**Figure 3.** Flow chart illustrating the procedure from raw PWS data to interpolated precipitation grids.

**Table 1.** Statistics of three Netatmo stations (N07, N10, N11) compared to a Pluvio weighing gauge for April to October 2019 at the IWS Meteorological Station for different temporal aggregations.

| | 1h | | | | 6h | | | | 24h | | | |
|---|---|---|---|---|---|---|---|---|---|---|---|---|
| | Pluvio | N07 | N10 | N11 | Pluvio | N07 | N10 | N11 | Pluvio | N07 | N10 | N11 |
| $p_0$ [-] | 0.92 | 0.84 | 0.94 | 0.91 | 0.82 | 0.75 | 0.84 | 0.82 | 0.59 | 0.56 | 0.65 | 0.59 |
| mean [mm] | 1.24 | 1.46 | 1.80 | 1.41 | 3.46 | 4.04 | 4.24 | 3.89 | 5.78 | 7.28 | 7.51 | 7.02 |
| standard deviation [mm] | 2.15 | 2.52 | 4.49 | 2.52 | 4.86 | 5.77 | 7.55 | 5.71 | 8.46 | 10.49 | 11.52 | 10.33 |
| 25th percentile [mm] | 0.18 | 0.20 | 0.10 | 0.20 | 0.39 | 0.33 | 0.30 | 0.40 | 0.48 | 0.63 | 0.58 | 0.58 |
| 50th percentile [mm] | 0.51 | 0.71 | 0.50 | 0.61 | 1.49 | 1.41 | 0.91 | 1.21 | 2.36 | 2.78 | 1.62 | 2.58 |
| 75th percentile [mm] | 1.34 | 1.72 | 1.41 | 1.52 | 4.60 | 5.33 | 4.14 | 4.95 | 7.82 | 9.87 | 11.26 | 9.95 |
| maximum [mm] | 19.84 | 22.62 | 44.74 | 22.22 | 23.28 | 28.58 | 44.74 | 27.98 | 45.62 | 55.55 | 56.16 | 55.55 |

All statistics except for the $p_0$ values are based on non-0 values. $p_0$ is the non-exceedance probability of precipitation < 0.1 mm.

## 4  Application and Results

The section describing the application of the methodology is divided into three parts. First the rationale of the assumptions is investigated. In a second step, the methodology is applied on a large number of intense precipitation events on different temporal aggregations using a cross validation approach. This allows for an objective judgement of the applicability of the results. Finally, the results of the interpolation on a regular grid are shown and compared.

### 4.1  Justification of the methods

For a direct comparison between the secondary rain gauges and devices from the primary network, three Netatmo rain gauges were installed next to a Pluvio[2] weighing rain gauge (the same type as regularly used by the DWD) at the Institute for Modelling Hydraulic and Environmental Systems' (IWS) own weather station on the Campus of the University of Stuttgart. With this data from 15 May to 15 October 2019 a direct comparison between the different devices used in the primary and secondary network was possible.

Table 1 shows statistics of the three devices compared to those of the reference station. The secondary stations overestimated precipitation amounts by about 20 %. It can be observed that the differences between the reference and the Netatmo gauge are not linear, hence a data correction of the secondary gauges using a linear scaling factor is not sufficient. Furthermore, the maximum in the sub-daily aggregations from N10 shows an outlier. This was caused by an interrupted connection between the rain sensor and the base station. In this case, the total sum of precipitation over a longer time period was transferred at once (i.e. in one single measurement interval) when the connection was established again. Such transmissions errors lead to outliers which falsify the results. Figure 4 shows scatter plots of hourly rainfall data and the corresponding percentiles from these three Netatmo gauges and the reference station. The occurrence of high values and percentiles is similar for the primary and the secondary devices. The Netatmo station N10 however deviates substantially from the other measurements in the quantile

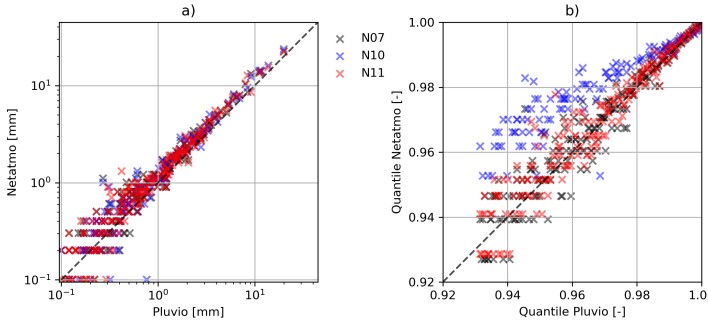

**Figure 4.** Scatter plot showing a) the hourly rainfall values (axes log-scaled) and b) the corresponding upper percentiles > 0.92 (right) between the Pluvio[2] weighing gauge and three Netatmo gauges (N07, N10, N11) at the IWS Meteorological Station.

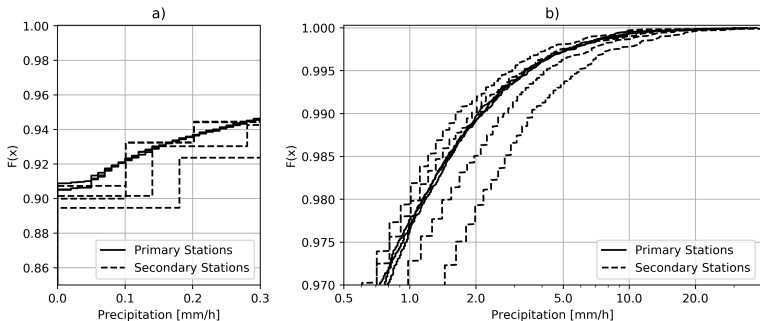

**Figure 5.** Probability of no precipitation (a) and the upper part of the empirical distribution functions (b) for three primary stations (solid lines) and four secondary stations (dashed lines) from a small area in the city of Reutlingen based on a sample size of 15,990 data pairs (hourly precipitation). The distance between the primary stations is between 5.5 to 9 km and the distances of the secondary stations to the next primary stations range from 1 to 3 km.

plot (Fig. 4b) which also points to data transmissions errors where the station failed to transmit data during rain events. The indicator filtering procedure (IBF) can identify such problems effectively.

The secondary measurement devices can also have very different biases depending on where and how they are installed. This can be seen by comparing the distribution functions of hourly precipitation data from nearby primary and secondary stations in the same area. Figure 5 shows the empirical distribution functions of three primary and four secondary stations in the city of Reutlingen. While the distribution functions of the primary network are nearly identical, those of the nearest secondary stations vary strongly. Some overestimate and others underestimate the amounts significantly. This example supports the concept of the paper, namely that secondary data require filtering and data transformations before use. While the distributions differ, the probability of no precipitation $p_0$ (defined as precipitation < 0.1 mm) ranges from 0.90 to 0.91 and is thus very similar for both types of stations indicating that the occurrence of precipitation can be well detected by the secondary network.

## 4.2 Application of the filters

Indicator correlations were calculated for different temporal aggregations and for a large number of different $\alpha$ values in the range between 95 and 99 %. Figure 6 shows the indicator correlations for one hour aggregation and the 99 % quantile using pairs of observations of the primary-primary and the primary and secondary network as a function of station distance. The indicator correlations of the pairs of the primary network show relatively high values and a slow decrease with increasing distance. In contrast, if the indicator correlations are calculated using pairs with one location corresponding to the primary and one to the secondary network the scatter increased substantially. Secondary stations for which the indicator correlations are very small in the sense of equation (4) are considered as unreliable and are removed from further processing. A relatively large distance tolerance was used as the density of the primary stations is much lower than the density of the secondary stations. On the right panel the indicator correlations corresponding to the remaining secondary stations shows a similar spatial behaviour as the primary network. In our case, 2462 of the originally available 3082 stations remained with a time series length of more than two months. After applying the IBF filter, a set of 862 (35 %) PWS remained. This is a relatively small fraction of the total number of secondary stations, but note that the shortest records were removed and low correlations may occur as a consequence of short observation periods. In the future with increasing number of measurements some of these stations may be reconsidered.

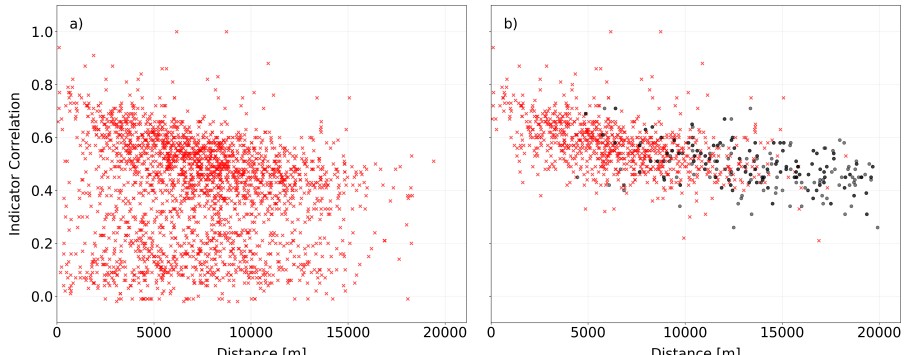

**Figure 6.** Indicator correlations for 1h temporal resolution and $\alpha = 0.99$ between the secondary network and the nearest primary network stations before (left) and after (right) applying the IBF (red crosses). The black dots refer to the indicator correlation between the primary network stations.

The effect of the IBF was checked by calculating the rank correlations between pairs of primary and PWS stations with a distance below 2,500m. Figure 7 shows that the removed PWS have a low rank correlation to their primary neighbours, while for the accepted ones the majority of the rank correlations is high. These high rank correlations support the rank based hypothesis formulated in equation (1).

The EBF was applied for each event individually. The number of discarded secondary stations is this study varied from event to event and was on average around 5 %.

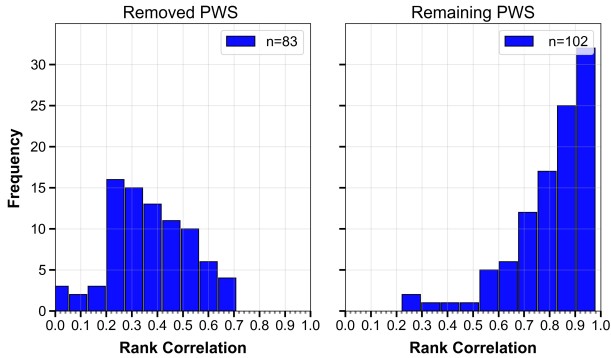

**Figure 7.** Histograms of the rank correlations between primary stations and PWS for pairs with a distance less than 2,500m. The left panel shows the rank correlations for the stations removed by the filter, the right panel for those which were accepted.

### 4.3 Bias correction

The bias correction method is illustrated using the example shown in Figure 8. For simplicity, 4 primary stations at the corners of a square and the secondary station in the center of this square are considered. This configuration ensures that the OK weights of the primary station with respect to the secondary station are all equal to $1/4$ independently of the variogram. The observed precipitation amounts at the corner stations are 3.1, 1.8, 3.0 and 2.1 mm for a selected event. The secondary station in the centre recorded 1.7 mm of rainfall. This corresponds to the 0.99 non-exceedance probability of precipitation for the specific secondary station. The precipitation quantiles at the primary stations corresponding to the 0.99 probability are 3.2, 3.5, 3.1 and 3.0 mm. Interpolation of these values gives 3.2 mm which is the value assigned to the secondary station instead of the value of 1.7 mm. This value is greater than all the four primary observations. The reason for this is that the primary observations all correspond to lower percentiles. Note that the interpolation of the primary values corresponding to the event for the secondary observation location would be 2.5 mm.

The bias in the PWS observations can be recognized by investigating data with higher temporal aggregation. The comparison of monthly or seasonal precipitation amounts primary stations and PWS reveals whether there is a systematic difference or not. As monthly or seasonal precipitation can be well interpolated by using primary stations only (temporal aggregation increases the quality of interpolation (Bárdossy and Pegram, 2013)), this comparison provides a good indication of bias. The difference between the interpolated and the PWS aggregations is different from PWS to PWS and often exceeds 20 %. Both positive and negative deviations occur. This points out that bias correction has to be done for each station separately.

### 4.4 Cross validation results

As there is no ground truth available the quality of the procedure had to be tested by comparing omitted observations and their estimates obtained after the application of the method.

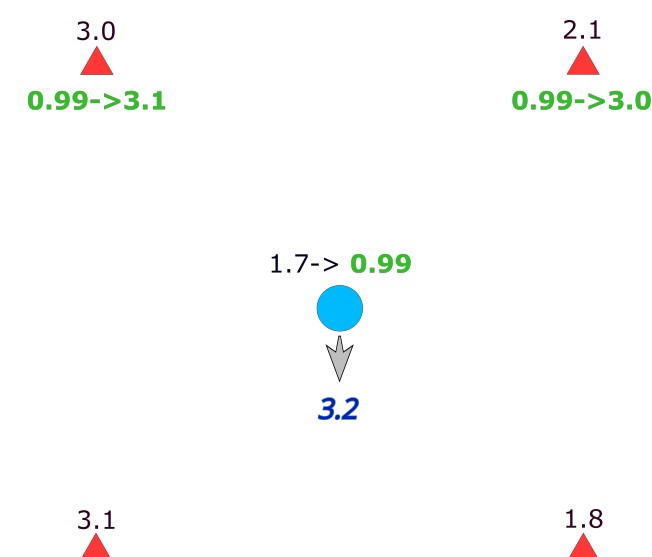

**Figure 8.** Example for transformation and bias correction of precipitation amounts at a secondary station.

**Table 2.** Statistics of the selected intense precipitation events based on the primary network.

| Temporal resolution | 1 hour | 3 hours | 6 hours | 12 hours | 24 hours |
|---|---|---|---|---|---|
| Number of intense events | 185 | 190 | 190 | 195 | 195 |
| Events between October-March | 1 | 16 | 29 | 48 | 57 |
| Events between April-September | 184 | 174 | 161 | 147 | 138 |
| Minimum of the maxima [mm] | 28.01 | 31.2 | 33.35 | 34.9 | 35.5 |
| Maximum of the maxima [mm] | 122.3 | 158.2 | 158.4 | 160 | 210.3 |
| $p_0$ (mean of all stations and events) | 0.9 | 0.84 | 0.77 | 0.68 | 0.55 |

$p_0$ is defined here as precipitation <0.1mm

The cross validation was carried out for a set of different temporal aggregations $\Delta t$ and a set of selected events. Only times with intense precipitation were selected . Table 2 shows some characteristics of the selected events. For short time periods nearly all events were from the summer season, while for higher aggregation the number of winter season events increased, but their portion remained below 30 %.

The improvement obtained through the use of secondary data is demonstrated using a cross validation procedure. The primary network is randomly split into 10 subsets of 10 or 11 stations each. The data of each of these subsets was removed and subsequently interpolated using two different configurations of the data used, namely a) only other primary network stations (Reference 1) and b) using the other primary and the secondary network stations (Reference 2). For the latter case, the interpolations were carried out using the primary station data and the following configurations:

- C1: All secondary stations

- C2: Secondary stations remaining after the application of the IBF

- C3: Secondary stations remaining after application of the IBF and the EBF

- C4: Secondary stations remaining after application of the IBF and the EBF and considering uncertainty (KU)

The results were compared to the observations of the removed stations. The comparison was done for each location using all time steps and at each time step using all locations. Different measures including those introduced in Bárdossy and Pegram (2013) were used to compare the different interpolations. The results were evaluated for each temporal aggregation.

First, the measured and interpolated values were compared for each individual station and the Pearson ($r$) and Spearman correlations ($r_S$) of the observed and interpolated series were calculated. Table 3 shows the results for the different configurations used for the interpolation.

**Table 3.** Percentage of the stations with improved temporal correlation (compared to interpolation using primary stations only) for the configurations C1-C4.

| Temporal aggregation | 1 hour | | 3 hours | | 6 hours | | 12 hours | | 24 hours | |
|---|---|---|---|---|---|---|---|---|---|---|
| Number of events | 185 | | 190 | | 190 | | 195 | | 195 | |
| Correlation measure | $r$ | $r_S$ | $r$ | $r_S$ | $r$ | $r_S$ | $r$ | $r_S$ | $r$ | $r_S$ |
| C1: Primary and all secondary without filter and OK | 60 | 68 | 40 | 57 | 31 | 49 | 22 | 34 | 17 | 32 |
| C2: Primary and secondary using IBF and OK | 81 | 91 | 75 | 90 | 73 | 90 | 64 | 84 | 52 | 81 |
| C3: Primary and secondary using IBF, EBF and OK | 81 | 92 | 75 | 93 | 73 | 92 | 69 | 92 | 56 | 87 |
| C4: Primary and secondary using IBF, EBF and KU | 81 | 92 | 75 | 92 | 74 | 91 | 70 | 91 | 56 | 86 |

$r$ Pearson correlation, $r_S$ Spearman correlation.

There is no improvement if no filter is applied - except a very slight improvement for 1 hour durations. This is mainly due to the better identification of the wet and dry areas. The use of the filters (and the subsequent transformation of the precipitation values) leads to an improvement of the estimation - the IBF being the most important. The spatial filter further improves the correlation while the additional consideration of the uncertainty of the corrected values at the secondary network resulted in a marginal improvement for the selected events. As the secondary stations are not uniformly distributed over the investigated domain the gain of using them is also not uniform. Highest improvements were achieved in and near urban areas with a high density of secondary stations, less improvement was achieved in forested areas with few secondary stations.

The measured and interpolated results were also compared for each event in space and the correlations between the observed and the interpolated spatial patterns were calculated as well. Table 4 shows the frequency of improvements for the different configurations C1 to C4 used for the interpolation.

The use of secondary stations leads to a frequent improvement of the spatial interpolation even in the unfiltered case. The reason for this is that the spatial pattern is reasonably well captured by the secondary network. With increasing temporal

**Table 4.** Percentage of the stations with improved spatial correlation (compared to interpolation using primary stations only) for the configurations C1-C4 ( $r$ Pearson correlation, $r_S$ Spearman correlation)

| Temporal aggregation | 1 hour | | 3 hours | | 6 hours | | 12 hours | | 24 hours | |
|---|---|---|---|---|---|---|---|---|---|---|
| Number of events | 185 | | 190 | | 190 | | 195 | | 195 | |
| Correlation measure | $r$ | $r_S$ | $r$ | $r_S$ | $r$ | $r_S$ | $r$ | $r_S$ | $r$ | $r_S$ |
| C1: Primary and all secondary without filter and OK | 83 | 68 | 72 | 52 | 63 | 49 | 53 | 49 | 49 | 46 |
| C2: Primary and secondary using IBF and OK | 96 | 97 | 90 | 93 | 90 | 93 | 84 | 89 | 80 | 85 |
| C3: Primary and secondary using IBF, EBF and OK | 96 | 97 | 92 | 94 | 93 | 94 | 89 | 92 | 84 | 89 |
| C4: Primary and secondary using IBF, EBF and KU | 93 | 94 | 90 | 92 | 90 | 93 | 84 | 89 | 80 | 87 |

aggregation the improvement disappears as the role of the bias increases due to the decreasing number of data which can be used for bias correction. As in the case of the temporal evaluation the IBF (and the subsequent transformation of the precipitation values) leads to the highest improvement. The EBF plays a marginal role, and the consideration of the uncertainty leads to a slight reduction of the quality of the spatial pattern. The improvement is smaller for higher temporal aggregations.

Kriging with uncertainty did not improve the results.

Finally, all results were compared in both space and time. Here the root mean squared error (RMSE) was calculated for all events and control stations. Table 5 shows the results for the different configurations used for the interpolation.

**Table 5.** RMSE (mm) for all stations and events.

| Temporal aggregation | 1 hour | 3 hours | 6 hours | 12 hours | 24 hours |
|---|---|---|---|---|---|
| Number of events | 185 | 190 | 190 | 195 | 195 |
| C0: Primary stations only and OK (Reference) | 5.97 | 6.97 | 7.34 | 7.71 | 8.35 |
| C1: Primary and all secondary without filter and OK | 6.21 | 44.79 | 18.43 | 10.01 | 24.16 |
| C2: Primary and secondary using IBF and OK | 4.83 | 6.05 | 6.61 | 7.33 | 8.29 |
| C3: Primary and secondary using IBF, EBF and OK | 4.84 | 6.07 | 6.58 | 7.19 | 8.12 |
| C4: Primary and secondary using IBF, EBF and KU | 4.82 | 6.02 | 6.53 | 7.15 | 8.08 |

The improvement using the filters is high for each aggregation. The IBF is important to improve interpolation quality. The EBF and the consideration of the uncertainty of the secondary stations are of minor importance. The improvement is the largest

for the shortest aggregation (1 hour) where the RMSE decreased by 20 % and the smallest for the 24 hours aggregation with an improvement of 4 %. This deterioration is caused by the decreasing spatial variability of precipitation at higher temporal aggregations. The processes that lead to long lasting precipitation are predominantly accompanied by a more even distribution of precipitation in space and time. The use of KU for interpolation resulted only in a minor improvement. Nevertheless, it is reasonable to assign lower weights to the less reliable PWS data. In order to check whether the selection of the events led to

this result a cross validation for all 1 hour time steps during the period from April to October 2019 (5,136 time steps) was carried out. The results are shown in Table 6. In this case, OK with secondary data did not lead to an improvement. This is

mainly caused by the irregular spatial distribution of the PWS. Stations located very close to each other can cause instabilities in the solution of the Kriging equations leading to high positive and negative weights. Introducing a small random error (1 %) to the PWS stabilizes the solution and leads to an improvement of the interpolation. The more realistic random error of 10 % further improves the results.

**Table 6.** RMSE (mm) and correlations for all stations for all time steps (5136) between April and October 2019 for OK and KU with different error assumptions for 1h aggregation.

| Interpolation method | RMSE | Correlation | Rank correlation |
|---|---|---|---|
| Primary stations OK | 0.331 | 0.640 | 0.443 |
| Primary and PWS OK | 3.862 | 0.644 | 0.402 |
| Primary and PWS EK (1% error) | 0.314 | 0.759 | 0.578 |
| Primary and PWS EK (10% error) | 0.158 | 0.809 | 0.631 |

Note that the use of the filtered and bias corrected secondary stations improves the interpolation quality even for other interpolation methods. Table 7 shows the results for the 185 events with 1 hour aggregation. One can observe that KU gives the best results, but the simple interpolations Nearest Neighbour or Inverse Distance also lead to better results than using primary stations only. The poor performance of the Co-Kriging is surprising. For this study we used the observations from the secondary stations as co-variable. The linear relationship which is supposed to exist between the investigated variable (precipitation) and the secondary variable (precipitation measured at PWS) for the application of Co-Kriging may not be appropriate for this combination of variables. Considering the ranks of the secondary observations or other transformed values as co-variables may improve the Co-Kriging results, but this is not the primary topic of this paper.

**Table 7.** Bias and RMSE (mm) for all stations and events for different interpolation methods for 1h aggregation.

| Interpolation method | Bias | RMSE |
|---|---|---|
| Ordinary Kriging primary data only | 0.05 | 5.97 |
| Kriging with uncertainty primary + PWS | 0.50 | 4.82 |
| Nearest Neighbour primary + PWS | 0.89 | 5.06 |
| Inverse Distance primary + PWS | 0.89 | 5.27 |
| Co-Kriging primary + PWS | 0.16 | 5.32 |

## 4.5 Selected Events

As the cross validation results were showing improvements, the data transformations and subsequent interpolations were carried out for all selected events. As an illustration four selected events are shown and discussed here.

The first example (Fig. 9) shows the results of the interpolation of a 1 hour aggregated precipitation amount for the time period from 15:00 to 16:00 on June 11, 2018. For this event, 531 out of 862 PWS had valid data (i.e. not NaN) from which 476 remained after the EBF. The top panels of this figure show three different precipitation interpolations for this event:

a) using the combination of the two station networks after application of the filters and transformation of the secondary data

b) using the primary network only

c) using all raw unfiltered and uncorrected data from the secondary network only

The panels in the bottom row of Figure 9 show d) the difference between a) and b), and e) the difference between c) and b). The three images a) to c) are similar in their rough structure, but there are important differences in the details. The interpolation using the primary network leads to a relatively smooth surface. The unfiltered secondary station based interpolation is highly variable and shows distinct patterns such as small dry and wet areas. The combination after filtering and transformation is more detailed than the primary interpolation, and in some regions these differences are high. The map of the difference between the primary and the secondary station based interpolation (Fig. 9 e) shows large regions of underestimation and overestimation by the secondary network. The differences between the primary and the filtered interpolations using transformed secondary data in panel d) is much smaller but in some regions the differences are still quite large, e.g. in the north-eastern part of the study area. In both cases, negative and positive differences occur. Note that for this data the cross validation based on the primary observations showed an improvement of $r$ from 0.36 to 0.77, of $r_S$ from 0.55 to 0.76 and a reduction of the RMSE from 12.5 mm to 8.2 mm.

Figure 10 shows the distributions of the cross validation errors for the different interpolations for this event. This is a typical case where all methods yield unbiased results. The use of unfiltered and uncorrected secondary observations (C1) shows the highest variance, followed by the interpolation using only primary observations (C0). The other three methods (C2-C4) have very similar results with significantly lower variance.

Another interpolated 1 hour accumulation corresponding to 17:00 to 18:00 on September 6, 2018 is shown in Figure 11. For this event, from the 862 PWS remaining after the IBF, 576 PWS had available data from which 513 remained after the EBF. These pictures show a similar behaviour to those obtained for June 11 (Fig. 9). Here, a high local rainfall in the southern central part of the study area was obviously not captured by the secondary network, leading to a large local underestimation in panel e). Furthermore, a larger area with precipitation in the primary network in the northern central in panel b) is significantly reduced in size by the rainfall/no-rainfall information from the secondary network in panel c). For this case, the cross validation based on the primary observations showed an improvement of $r$ from 0.61 to 0.86, of $r_S$ from 0.59 to 0.72 and a reduction of the RMSE from 5.65 mm to 3.75 mm.

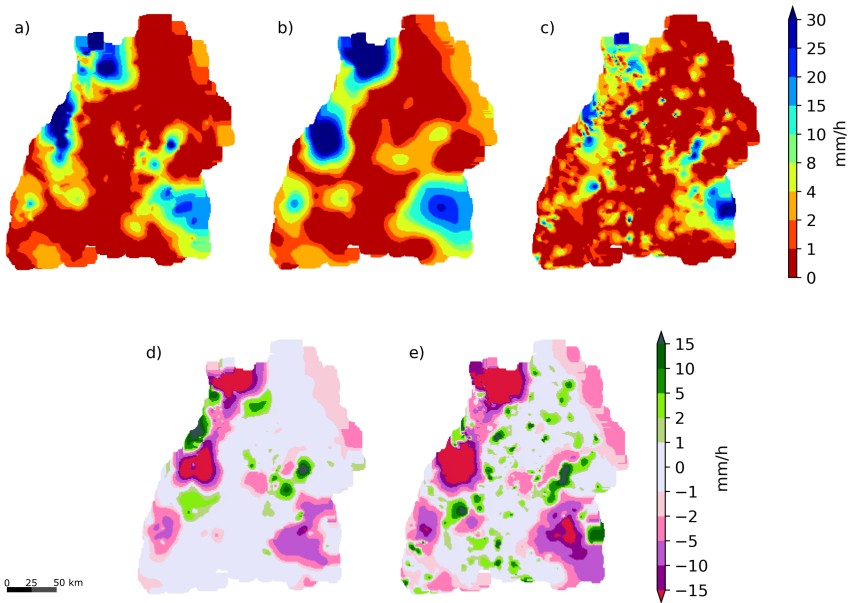

**Figure 9.** Interpolated precipitation for the time period 15:00 to 16:00 on June 11, 2018 (upper panel), and the differences between primary and combination, and primary and secondary data based interpolations. Panel a) shows the result after applying the filtering, b) the interpolation from the primary network and c) the one from the secondary network. Panels d) and e) depict the differences between a) and b) and c) and b) respectively.

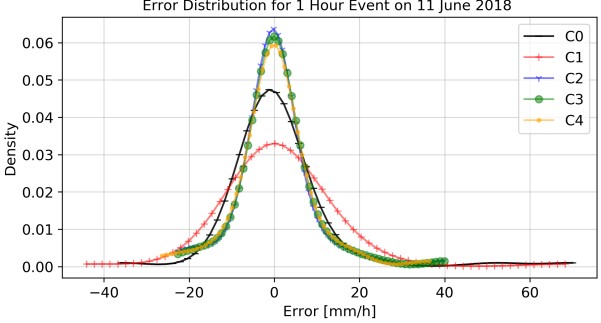

**Figure 10.** Distribution of the cross validation errors for the time period 15:00 to 16:00 on June 11, 2018 for the five interpolation methods: C0: using primary stations only and OK, C1: Primary and all secondary without filter and OK, C2: Primary and secondary using IBF and OK, C3: Primary and secondary using IBF, EBF and OK, C4: Primary and secondary using IBF, EBF and KU.

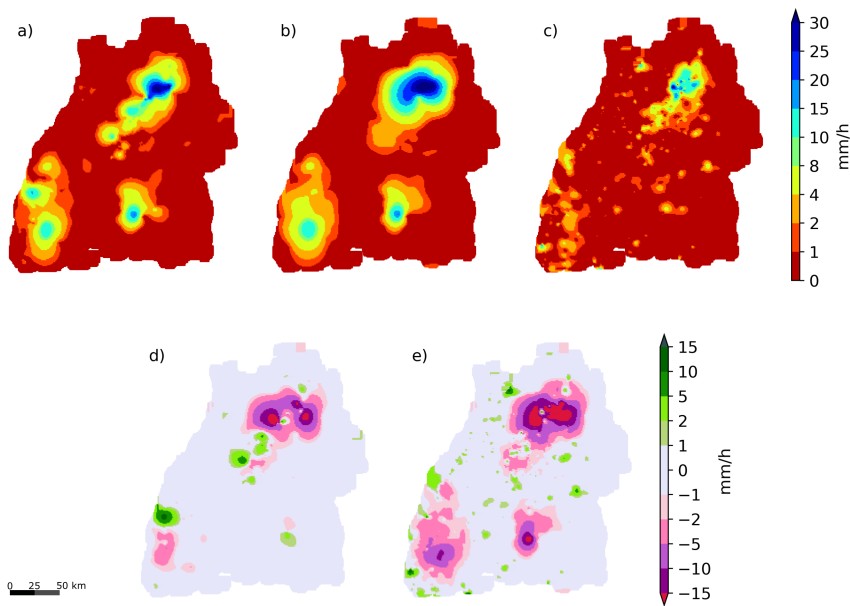

**Figure 11.** Interpolated precipitation for the time period 17:00 to 18:00 on September 6, 2018 (upper panel) and the differences between primary and combination and primary and secondary data based interpolations. Panel a) shows the result after applying the filtering, b) the interpolation from the primary network and c) the one from the secondary network. Panels d) and e) depict the differences between a) and b) and c) and b) respectively.

The following two case studies show two interpolation examples for 24 hours which was the highest temporal aggregation in this study. Figure 12 shows the maps corresponding to the precipitation of 0:00 to 24:00 on May 14, 2018. For this event, 515 PWS valid stations remained. This number was reduced to 499 after the EBF. The behaviour of the interpolations is similar to the 1 hour cases shown above, the unfiltered and untransformed secondary interpolation is irregular and shows a systematic underestimation. Due to the higher temporal aggregation, the local differences are less contrasting as in the case of hourly maps. The combination contains more details and the transition between high and low intensity precipitation is more complex. The difference between the primary (panel b) and the combination based interpolation in panel a) is relatively smaller than for the 1 hour aggregations. This is caused by the reduction of the variability with increasing number of observations. Note that for this event the cross validation based on the primary observations showed an improvement of $r$ from 0.57 to 0.8, of $r_S$ from 0.57 to 0.82 and a reduction of the RMSE from 15.99 mm to 13.61 mm.

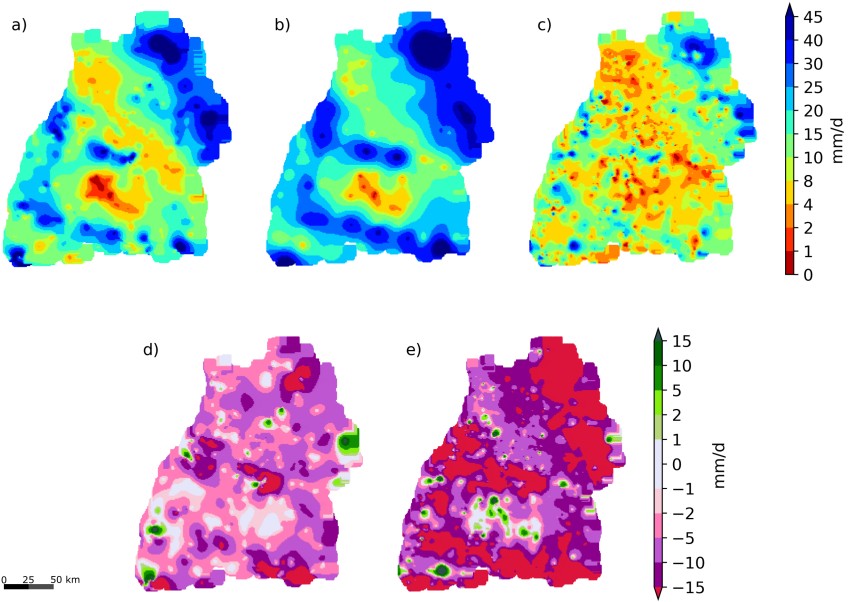

**Figure 12.** Interpolated precipitation for the time period for a 24 hour event from 0:00 to 24:00 on May 14, 2018 (upper panel) and the differences between primary and combination and primary and secondary data based interpolations. Panel a) shows the result after applying the filtering, b) the interpolation from the primary network and c) the one from the secondary network. Panels d) and e) depict the differences between a) and b) and c) and b) respectively.

Another interesting 24 hour event which was recorded on July 28, 2019 is shown in figure 13. For this event, 734 valid PWS remained from IBF and 703 after EBF. The map based on the raw secondary data in panel c) shows very scattered intense rainfall. The combination of the primary and secondary observations changes the structure and the connectivity of these area with intense precipitation. The cross validation for this event showed an improvement of $r$ from 0.32 to 0.75, of $r_S$ from 0.42 to 0.77 and a reduction of the RMSE from 14.77 mm to 10.21 mm.

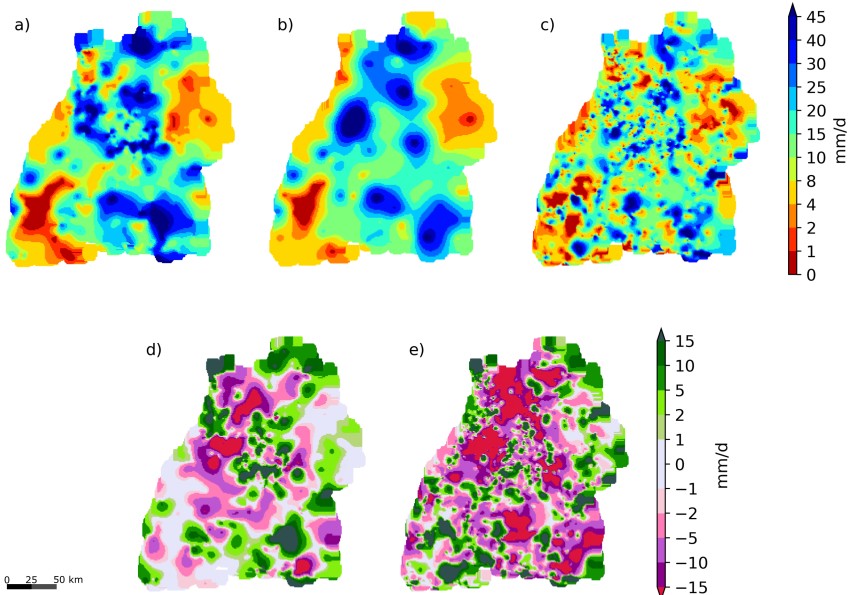

**Figure 13.** Interpolated precipitation for the time period for a 24h event from 0:00 to 24:00 on July 28, 2019 (upper panel) and the differences between primary and combination and primary and secondary data based interpolations. Panel a) shows the result after applying the filtering, b) the interpolation from the primary network and c) the one from the secondary network. Panels d) and e) depict the differences between a) and b) and c) and b) respectively.

The results of the filtering algorithm for the other events show a similar behaviour. The differences between primary and combined interpolation can be both positive and negative for all temporal aggregations. In general, the secondary network provides more spatial details, which could be very important for hydrological modelling of meso-scale catchments.

Figure 14 shows the distributions of the cross validation errors for the different interpolations for this event. The results are different from the case presented in Figure 10. In this case all methods are slightly biased. The interpolation using only primary observations (C0) shows the highest bias and variance. In this case, the use of unfiltered and uncorrected secondary observations (C1) yields a lower bias and a lower variance. The other three methods (C2-C4) have very similar results with significantly lower variance.

**5    Discussion**

The use of observations from such PWS networks has the potential to improve the quality of precipitation estimation. However, the results from this study as well as the ones from de Vos et al. (2019) show that it is necessary to check the data quality from PWS precipitation records and to discard erroneous measurements before further using these data.

     There are already several approaches to use the precipitation data from PWS (e.g. Chen et al., 2018; Cifelli et al., 2005), but 450    they are generally based on daily data an simple QC approaches. Studies using more sophisticated QC workflows for hourly or

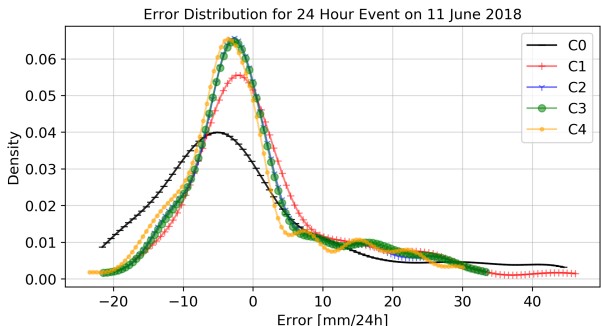

**Figure 14.** Distribution of the cross validation errors for the 24h event from 0:00 to 24:00 on July 28 2018, for the five interpolation methods: C0: using primary stations only and OK, C1: Primary and all secondary without filter and OK, C2: Primary and secondary using IBF and OK, C3: Primary and secondary using IBF, EBF and OK, C4: Primary and secondary using IBF, EBF and KU.

sub-hourly precipitation data from PWS are still limited. The approach presented by de Vos et al. (2019) uses a comparison of the data with those of the nearby stations to remove unreasonable values, a separate procedure to identify and remove false zeros and another filter to find unreasonably high values. Subsequently, the bias is corrected by comparing past local observations to a high quality merged radar and point observation product. The bias correction is performed uniformly in neighbourhoods.

Finally, another filter using correlations of time series serves to remove remaining suspicious data. In the study presented here, a geostatistical method combined with rank statistics was developed. One of the main difference to the method presented by de Vos et al. (2019) is that a set of trustworthy precipitation data (primary stations) is required for the rank correlation and the bias correction. First, PWS which have indicator time series with low correlations compared to the primary network are removed. The remaining secondary stations are tested for each event separately using OK in a cross validation mode. Finally

the data are bias corrected using interpolated quantiles of the primary observations. This is an important aspect, since PWS that are close to each other do not necessarily have a similar bias. Examples from the Reutlingen data show that positive and negative biases can occur at neighbouring PWS. The bias correction in this study does not use simultaneous observations of the primary and the PWS stations, but instead is based on their distributions. A detailed cross-validation of different filter combinations and temporal aggregations shows that the IBF is the most important step and yields the highest improvement in

interpolation quality, whereas the EBF and bias correction only have a minor contribution. Furthermore, the performance of the presented method is better at smaller temporal aggregations. The applied filters in this study may be conservative by rejecting more stations than absolutely needed, but this proved to be useful in order to obtain robust results. The length of times series from the current secondary network will increase and subsequently more observations which were currently discarded may also become useful. Furthermore, it can be expected that the number of secondary stations will continue to increase, thus one can

expect further improvements of the quality of precipitation maps for all temporal aggregations. Overall, the use of secondary stations after filtering and data transformation improves the results of interpolation for other possible interpolation methods, such as nearest neighbour or inverse distance weighting. However, in this study these methods yielded worse results than OK.

An advantage of the KU interpolation method is that a combination of different measurements, such as radar estimates or commercial microwave links which are based indirect information can be accommodated in the same framework. By using KU for interpolation, the weights for data from secondary networks can be reduced to account for the higher uncertainty of these data. Other procedures for the efficient use of secondary data may also be considered. Specifically, the interpolation of precipitation amounts with Co-Kriging using non-collocated observations (Clark et al., 1989) using percentiles $P_{\Delta t}(y_j, t)$ as co-variates (eq. 5) or Quantile Kriging (QK) (Lebrenz and Bárdossy, 2019) may lead to better results. However QK has to be modified due to the large number of zeros occurring for short temporal aggregations, for example by combining it with the approach developed by Bárdossy (2011).

A problem that affects both primary and PWS stations are errors caused by wind. In general, this has a major effect on precipitation measurements leading to a systematic undercatch. These effects might differ from station to station and cannot be corrected. The suggested methodology uses ranks and not the measured precipitation values of the PWS. Thus, the problem related to wind only affects the results if it changes the order of the precipitation measured at the same location. This order however is relatively stable for high precipitation values, as due to the skewness of the distribution the difference between the measured values is high.

## 6  Conclusions and Outlook

As precipitation uncertainty is possibly the most important factor for the uncertainty in rainfall/run-off modelling, the increasing number of online available private weather stations offers a possibility to increase the accuracy of precipitation estimation. Furthermore, the near real-time availability of the data of secondary networks may help to improve the quality of flood forecasts. In any case, a QC of these data is required since the use of raw data of the secondary network does not improve interpolation quality; on the contrary it often increases uncertainty. In this study, a geostatistical method combined with rank statistics was applied to combine data from primary and PWS networks. In particular:

 – An assumption on the rank stability of the PWS stations was introduced.

 – A new method to filter out erroneous PWS data based on indicator correlations was developed.

 – A second geostatistical filter to remove individual PWS observations was applied.

 – A rank statistics based bias correction was developed. The bias correction does not use simultaneous observations of the primary and the PWS stations, but instead is based on their distributions.

 – A Kriging interpolation with uncertain data was used (KU). This method allows for a down-weighting of the PWS stations and leads to an improvement of the interpolation quality.

This approach was tested on a set of observations and the improvement of the quality of interpolation was quantified. A detailed cross validation experiment showed that after QC and bias correction in a large number of cases interpolation quality was improved. This improvement is the biggest for hourly temporal aggregations with a reduction of the RMSE by 20 % ,

while for daily values the improvement is around 4 %. The results of this study in terms of improving the interpolation of precipitation are encouraging, but the authors believe that further improvements can be achieved. In this context, the following aspects would be of interest:

1.) In this study, the number of primary station was sufficient to improve the interpolation quality. However, it would be interesting to investigate which density of primary stations is necessary to improve the precipitation interpolation.

2.) For applying this approach to shorter time steps (e.g. 5 minutes for which the PWS data is available), the effect of advection would have to be taken into account. This requires further research.

3.) By applying a rather strict threshold of 5°C average daily temperature, many rainfall events were rejected. It would be conceivable to include the hourly temperature data from PWS in order to estimate whether a given precipitation event corresponds to rain or snow.

*Data availability.* The precipitation data was obtained from the Climate Data Center of the German Weather Service (https://opendata. dwd.de/climate_environment/CDC). The data from the Netamo stations was downloaded using the Netatmo API (https://dev.netatmo.com/ apidocumentation).

*Author contributions.* AB designed the study, AEH implemented the filtering algorithm for the study area. JS conducted the case studies in the chapter for the justification of the methods. All authors contributed to the writing, reviewing and editing of the manuscript.

*Competing interests.* The authors declare that they have no conflict of interest.

*Acknowledgements.* The authors would like to thank Lotte de Vos, Nadav Peleg, Mark Schleiss, Hannes Tomy-Müller an one anonymous reviewer for their time to provide constructive and comprehensive comments which helped to improve this manuscript. Furthermore, Faizan Anwar is acknowledged for his help with the computer codes for the assessment of the secondary data. This publication was supported by the Open Access Publishing Fund of the University of Stuttgart.

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
