# Peer review of "The use of personal weather station observations for improving precipitation estimation and interpolation"

_Hydrology and Earth System Sciences, 2020_

## Referee Comment (RC1) · Lotte de Vos (Referee) · 14 Feb 2020

**Review**

*Title: The use of personal weather station observation for improving precipitation estimation and interpolation*
*Author(s): András Bárdossy et al.*
*MS No.: hess-2020-42*
*MS Type: Research article*
*Reviewer: Dr L.W. de Vos*

**Summary**

In this paper rainfall observations from personal weather stations (PWSs) are used to describe the spatial interpolation of heavy rainfall events, while taking into consideration the typical unknown errors and biases of the observed rainfall estimations from PWSs. The authors present a novel approach to use PWS observations to better capture the variability of heavy rainfall events than a primary sensor network is able to. The method is evaluated on a substantial dataset of almost 200 intense events in Baden-Württenberg in Germany. The Netatmo PWS network can provide rainfall observations at densities that traditional operational sensor network typically lack. Making use of this data source for extreme events can therefore be highly valuable.

The paper is well-written with few grammatical errors and high-quality figures, and addresses an interesting problem (high resolution rainfall monitoring of heavy rainfall events) in a novel way. The paper would benefit from clarifying some choices and assumptions in the methodology, and describing the resulting limitations thereof. Also, some steps in the method could be presented in a more straightforward way. When these issues are addressed, the resulting paper will be highly useful and relevant.

**Major comments**

P2L42-49: In the introduction, the paper of de Vos et al. (2019) is discussed, which proposes a quality control (QC) methodology for PWS rainfall observations. This QC has been applied on a Netatmo dataset in the Amsterdam metropolitan area as well as a national dataset of the Netherlands. The PWS measurements were compared with a gauge-adjusted radar product before and after the QC was applied, in order to demonstrate the improved accuracy. This validation showed that the QC was successful in flagging intervals with errors, without the need for auxiliary data, and thereby reducing the bias, increasing the Pearson correlation coefficient and reducing the coefficient of variation, by excluding the flagged intervals (12% of the original dataset of a year in the Amsterdam metropolitan area).

The gauge-adjusted radar product in that study was merely used as a ground-truth, not as a vital part in the QC. There may have been some confusion as there is one parameter in that QC methodology, the *DBC*, which is a proxy value used in the QC to compensate the overall bias in the network, that was determined in an offline exercise using the gauge-adjusted radar product. It is also clearly mentioned in the paper that this *DBC* does not need to be estimated in that manner or at all, as proven by the accuracy improvement after QC was applied when *DBC* was chosen as 1 (i.e. no proxy was made at all), see de Vos et al. (2019) Table 2.

Therefore, the statement that the QC described by de Vos et al. (2019) is "based on combined official rain gauge and radar product" is faulty, and the statements regarding the limitations and uncertainties in radar QPE in this context are irrelevant. Moreover, the statement "the study by de Vos et al. (2019) does not provide a guideline on how to use the measurements of the PWS if no radar observations are available" is wrong, and the statement "data quality issues have to be overcome" in this line of reasoning is debatable. Also in the Discussion and conclusion section it is wrongly stated "...or relies on other data

sources as reference, such as precipitation estimates from weather radars which have an appropriate spatial and temporal resolution (de Vos et al. 2019)."

The study presented in this paper does not lose its relevance by the existence of a proven QC methodology that identifies erroneous PWS rainfall observations without auxiliary information, as it has merit in its specific focus on extreme precipitation events. Nevertheless, the previous studies that are described to reflect the current state of the art in this field need to be represented accurately.

The proposed method is interesting and promising, however there are some significant limitations due to the assumptions in the filters. It can be considered contradictory that the main perceived issue with the QC in previous work (mistakenly) is its dependence on another data source, while this methodology relies on the availability of another data source itself. The PWS are used as an addition to a high quality primary rain gauge network with long observation series in the study area of interest, measuring in high temporal resolution. Such a network may not be readily available everywhere, and this should be mentioned in the discussion more broadly than it is now.

Another important limitation of the applicability of this method is that the selection of stations to be included needs to be done over a considerable period for the high intensity based filtering to work. This assumes constant and steady PWS performance, and doesn't allow for changes in measurement accuracy in time (e.g. due to interference to the station or temporary blockage of the Netatmo rain gauge tipping bucket system). A station is either included or excluded in its totality, while previous research on Netatmo observations has shown that measurement accuracy can change drastically instantly for the better or worse. This should be included in the discussion as well, including the minimum required period over which the event should extend in order to reliably apply the method.

The paper is very limited in describing how the data is gathered from the Netatmo rain gauges, which measure approximately every 5 minutes. The unprocessed time series that can be collected with the Netatmo API do typically not have fixed time steps and can contain large data gaps. The paper is not clear on how these raw time series are processed into structured aggregated time series at 1, 3, 6, 12 and 24 hour time steps, but does mention in the evaluation of Netatmo data from the experimental set-up with a Pluvio sensor an error resulting from station connectivity. This error is difficult to understand without knowing the process that the authors have used.

**Minor comments**
- P1L18: "in situ sensors" -> "in-situ sensors"
- P1L19: "low cost personal weather stations" -> "low-cost personal weather stations"
- P1L20: "Netamto" -> "Netatmo"
- P2L39: "...a PWS rain gauge." –> "...three PWS rain gauges."
- P2L49: "real time flood forecasting" -> "real-time flood forecasting"
- P2L50: "two fold approach" -> "two-fold approach"
- In some number values the thousands are indicated with comma, in some cases not.
- Both "Figure", "Fig" and "Fig." is used in the text.
- P4L76: "time" -> "period"
- Caption Figure 1: "Map Of" -> "Map of"
- P4L77: "one can see that many stations have less than one year of observations" -> how does that follow (from figure 2 or elsewhere), and why is the proposed methodology not able to accommodate these stations?
- Section 2 would benefit from more quantitative descriptions of the measurement uncertainty of the sensors that are mentioned, e.g. from technical documentation of these sensors from the supplier.
- P5L96: "Since is..." -> "Since it is..."

- P5L103: "Note that *Y* is considered to be a random field, and thus methods like Co-Kriging or Kriging with an external drift are not applicable." -> the purpose of this statement in this context is not entirely clear to me.
- P5L105: probability (α) in Eq (1) needs to be explained more fully.
- Section 3.1 describes that a secondary station is flagged as suspicious if its indicator correlations with the nearest primary network points are below the lowest indicator correlation corresponding to the primary network for the same time steps and at the same separation distance. I can imagine that not all distances between secondary station and nearest primary network points equal a separation distance between two primary network stations exactly. Is then the nearest distance used? If so, what are the largest differences between separation distances? Or is the relationship between distance and correlation ($\rho$) described with a fitted relation (effectively a correlogram)? If so, what is then the meaning of "min" in Eq. (2)?
- P7L163: ".. due to unforeseen events (such as battery failure or transmission errors) at certain times they may deliver individual false values." -> How is the issue of data gaps in Netatmo time series addressed? Here it seems to be referred to as "false values", however it should be evident from the Netatmo time series that an observation was lacking (due to a long duration between the timestamps of two subsequent observations). I wonder if regarding these observations as zero observations and subsequently identifying them with a simple geostatistical outlier detection method is the best approach. The author's may refer to the station in total (not a certain period in observations), which due to battery failure or transmission errors is considered to be faulty. If that is the case, which fraction of the data should be missing for a station to be considered a geostatistical outlier? A later section (P9L224-229) hints at problems due to data gaps which resulted in a large outlier, but it's not clear if these cannot be avoided by looking at the timestamps of the PWS observations. More information on how the raw irregular Netatmo PWS datasets are converted to timeseries with fixed time steps would be very helpful.
- P7L174: "...which are due to temporary loss of connection between the rain gauge module and the Netatmo base stations." There are many other possible reasons for false zeroes. The QC in de Vos et al. (2019) has dedicated an entire module to those types of errors (see the red marked observations in the left graphs in Figure 1 of that paper for an indication how often these occur). Note that those FZ-errors are not related to station outage, as those intervals are already excluded at an earlier step in the analysis.
- P8L209: "judgment" -> "judgement"
- Table 1: I would be very interested to see if all three Netatmo stations have yielded observations every ~5 min, and if there were data gaps, how many and how long these were. I assume the stations were not calibrated and have the default tipping bucket volume of 0.101 mm. It says the statistics are based on non-0 values. Does that mean that both Pluvio and Netatmo station need to measure non-0 rainfall, or only the reference (Pluvio)? This should be specified in the footnote of the table. Here $p_0$ likely refers to probability of precipitation, which is only mentioned later (P10L235). It should be introduced earlier in the text, including the equation to calculation it. Also consider including other metrics in the table like RMSE and correlation using the Pluvio observations as ground truth to validate the Netatmo stations.
- Consider using a different symbol than $p_0$ for probability of precipitation (P10L235), as it is very similar to the symbol used for correlation ($\rho$).
- P9L218: "on" -> "one"
- Figure 4: why are the lines of the Secondary Stations stepped and the Primary Stations not?
- P11L244: "equations (2)" -> "Eq. (2)" ?
- Caption Figure 5 -> "Xes" -> "crosses"
- P11L257: "low intensity" -> "low-intensity"
- Table 2 caption: I assume that $p_0$ still refers to probability of precipitation. Is it then the fraction of intervals where precipitation is larger than 0.1 mm? In that case it makes more sense to change the text in the table from "<0.1 mm" to ">0.1 mm". Also, "(mean of all stations and events)" is not very clear in this context, please explain.
- P12L260: "Note the high portion of zeros" -> where can this portion be found? It doesn't seem to be provided in Table 2. Should this be portions of intervals where precipitation is <0.1 mm?

- Table 2: what was the procedure to select these events?
- P12L265: "the temporal filter" and "the event based spatial filter" probably refer to 3.1 and 3.3 respectively. It would be helpful to name those two filters explicitly in the method section and uses those names throughout .
- P12L274: "Pearson ($r$) and Spearman ($\rho$) correlation" -> up until now I would have assumed the correlation that was introduced in section 3.1 to be the Pearson correlation. However, as the symbol $\rho$ was used in that section, that was likely actually Spearman. Either way, it should be specified in section 3.1. Also, what is the motivation to evaluate two types of correlation?
- Section 4.2: It is explained that two references are constructed using cross validation. Reference 1 is constructed by interpolating the subsets with only primary network stations, and Reference 2 is constructed by interpolating the subsets with primary and secondary network stations. What is the reason for constructing two references? From their captions it seems that Table 3 and 4 are based on comparisons with Reference 1. Is Reference 2 used somewhere else?
- P14L315: "(Fig. 6 e))" -> avoid double brackets.
- P15L321: "on" -> "in"
- P17L333: "(panel b))" -> avoid double brackets.
- P17L334: "This is caused by the reduction of the variability with increasing number of observations" -> Is that true? Why would the variability of a rainfall event be dictated by the number of observations in space? It seems to refer to the more smooth rainfall patterns found at daily scales compared to hourly scales, but this phrasing is confusing.
- P19P370: "in higher" -> "at higher"
- P19P370: "point precipitation" -> "precipitation as a point value"
- P20L381: "real time availability" -> "real-time availability"
- P20L399: "This corresponds to the 0.99 non-exceedence probability of precipitation for the specific secondary station." -> how does this follow from the information that is provided? Or is this provided information?
- P20L400: "The precipitation quantiles at the primary stations corresponding to the 0.99 probability are 3.2, 3.5, 3.1 and 3.0 mm." -> how does this follow from the information that is provided? Or is this provided information?
- Some interesting additional literature to refer to could be: https://www.nat-hazards-earth-syst-sci.net/20/299/2020/nhess-20-299-2020.pdf on the use of Netatmo data for describing deep convection features. Also, the QC method https://github.com/metno/TITAN could be mentioned in addition to the QC method of Napoly et al. in the introduction. Finally, Chen et al. (2018) "Trust me, my neighbors say it's raining outside: Ensuring data trustworthiness for crowdsourced weather stations." is an example for quality estimation of PWS rainfall data from the Wundermap platform.

---

## Referee Comment (RC2) · Nadav Peleg (Referee) · 17 Feb 2020

In their paper, Bárdossy et al. suggest ways to improve rainfall estimation using personal weather stations (PWS). More precisely, they discuss QR procedures and suggest a method to merge rainfall information from trustable monitoring stations with PWS data. The paper is interesting and falls within the scope of HESS. Although none is critical, there are a few issues I would like the authors to consider before recommending the paper for publication.

1. The motivation to use PWS in rainfall estimation is quite clear and well written in the introduction. However, many studies suggest various stochastic and deterministic methods to blend/merge/interpolate different rainfall products, e.g. combining data from rain-gauges, weather radar and CML together. Why not applying an already established method to merge data from trustable rain-gauges with PWS? There should be a short explanation in the introduction of why a new merging method is needed.
2. Empirical distributions are used for all PWS. I was wondering if it wouldn't be more accurate to use a specific distribution instead. For example, the same distribution can be fitted to all the trustable stations (but with different parameters), and the parameters can be spatially interpolated to the PWS (and other) locations.
3. I agree that the examples presented in figures 6 to 8 cannot be evaluated against "true-rainfall" due to a lack of spatial information. That is why I believe that there is an added value in comparing the outcomes of the interpolation with data emerging from the weather radar composite in Germany. If you do not trust the radar QPE, there is no need to compare the actual rainfall intensities, but just to demonstrate that the interpolated rainfall fields can assist in revealing high-intensity rainfall features that are "hidden" when using the official rain-gauge network alone.
4. The potential to use PWS to generate rainfall fields at a minutes-scale is very appealing, especially for applications in urban hydrology. I see the potential in using PWS to simulate rainfall fields at high temporal-resolution, but in the presented study no sub-hourly examples are presented. It will be nice to see if the potential to interpolate the rainfall at high-resolution can be fulfilled and to discuss the limitations of the PWS and methods in going to such fine scales.

Some other minor points to consider are listed below.

L52. "… may be biased" –I guess it refers to rainfall intensities.

L64. 10-min?

Figure 2. It can be presented as Supplementary Material.

L98. "at short time steps" - 1-min? 5-min?

L102-103. "…thus methods like Co-Kriging or Kriging with an external drift are not applicable" - at this point in the text, some further explanation is needed to put this sentence in context.

L102. "is considered to be a random field" - Why? Reading further, this sentence is clear. But it is not clear at first reading.

L105. It should be mentioned in the text that alpha defines the percentile threshold. I assume it is subjectively defined?

Equation 1. I assume $Fu$ stands for distribution function? Please clarify in the text. In addition, there are two commas with empty space in the left term of the equation.

Equation 2. $k$ and $m$ are not defined in the text.

Section 3.2. Consider adding a flow chart to illustrate the steps described in this section.

L137-138. "For locations of the secondary network they have to be estimated via interpolation." Or using weather radar data... It should be mentioned, even if you chose here to go with a different alternative.

L145. and seasonality

Equation 5. I assume that $F$ refers to empirical distribution? This should be explained in the text.

L185. There is no section 3.4.2.

L218. one

L239. Isn't 95 percentile too low threshold if the goal is to attract the extreme rainfall intensities? Especially for the fine temporal resolution, for which I assume the sample size is quite large.

L397-400. Wouldn't it be more accurate to fit a specific distribution to each secondary station, based on parameters obtained from the primary stations around it?

---

## Referee Comment (RC3) · Anonymous Referee #3 · 2 Mar 2020

General comments:

The manuscript provides an interesting approach to error correct and incorporate information from personal weather stations into spatial interpolation of precipitation for different temporal resolutions. The methodology is clear and plausible. The manuscript is well written and concise. I have only some minor comments for improvement (see detailed comments).

Detailed comments:

1. Line 20: spelling error Netamto -> Netatmo

2. Figure 1: Red triangles are difficult see against brown elevations. Please consider changing colour, e.g. to black and bigger triangles

3. Lines 102-103: Why can multivariate methods like Co-Kriging not applied to random fields?

4. Lines 146-147ff: The sentence with quantiles and percentiles first caused some confusion to me. After reading several times I understood that the term "quantiles" is used here for precipitation values with certain non-exceedance probabilities (Eq. 5), which is common. But the term "percentiles" is used here for the non-exceedance probabilities (Eq. 4), which is not always common. Often, it also refers to the quantiles which divide the distribution into 100 equal portions. In order to avoid confusion, I would suggest beside giving equa-tion (4) also verbally to make clear that with percentiles the non-exceedance probability is referred to. Please, also make a comment on G(y) and F(x) if here empirical or theoretical distributions will be used.

5. Equation (5,6): It becomes not immediately clear which x(i) locations are related the y(j) location. Please, explain in the text and make a reference to Appendix A here.

6. Line 160: The estimate for y at time t can be bigger the observation at this time but cannot be bigger than the maximum observation for all times t at x, if an empirical distribution for F(x) is used. Please comment.

7. Line 205: Is there a reference available for KU?

8. Table 1: The definition of p0 is missing.

9. Lines 265ff: Please add interpolation methods OK or KU.

10. Line 272, Tables 3,4: I would suggest to name the errors "temporal error" and "spa-tial error" and repeat this in Tables 3 and 4. The terms temporal and spatial correlation in the tables might be misleading. These are correlations as performance measures to quantify the spatial and temporal errors.

11. Line 277: "There is no improvement ..." From Table 3 I see improvement for the different time aggregations between 17% – 60% of the stations?

12. Table 4: In the header I think it should read here "Percentage of time steps ..." not of stations, if the correlation is calculated for each time step using all stations?

13. Figures 6-9: In order to be consistent with the terms "under- und overestimation" of the secondary data based interpolation in comparison to the primary data based interpolation (as it is used here e.g. in Lines 323 and 331) I would suggest to change the differences in d) to a)-b) and e) to c)-b).
* * *

---

## Referee Comment (RC4) · Hannes Müller-Thomy (Referee) · 5 Mar 2020

General comments:

The manuscript introduces a multi-level data quality check for rainfall data measured by citizens and quantifies the improvement in time and space if the data is used for interpolation additionally to a conservative network. This kind of rainfall data is analysed for Germany for the first time to my knowledge and shows a high potential for further applications. Hence, this topic is of broad interest for the hydrological scientific community and suitable for a publication in HESS. The paper is well-written, with a clear and intuitive structure. I have no major comments, but a number of issues that should be solved before the manuscript should be published.

Specific comments:

L26-28 The short periods of available radar data should be mentioned in this context as well.

L77-79 It would be helpful if the authors are more concise regarding the number os PWS stations finally used in the study. To enable a transfer of the applied methods the authors should provide some information, which minimum time series length was chosen for the secondary time series and how was it chosen?

L85 From the first paragraph in Section 3 it sounds as only the two data quality filters will be explained. I suggest to provide a brief overview of all subsections at the beginning of Section 3 and an explanation, how they are interacting.

L102 Maybe the authors should explain briefly why they consider Y as a random field.

L120-123 The chosen criterion sounds reasonable. I'm wondering if an exclusion for too high correlations has to be applied as well. Later in Fig. 5 indicator correlations of 1 are shown for interstation distances of 10 km, which is way higher than from the primary network. Maybe the authors can report if an upper limit is required or not when working with the data as a result from their data analysis.

Also, I'm struggling with the final decision if a secondary time series remains in the potential useful data set or not. As far as I understand it a time series is "flagged as suspicious" if it does not meet the criterion in Eq. 2. That means the time series will be sorted out. Since the procedure is repeated for several $\alpha$ and $\Delta t$, I imagine the highest exclusion rate will be found for high values of $\alpha$. Is a flagging for only one of the analysed values of $\alpha$ enough for an exclusion of that time series? Which values of $\alpha$ have been applied and what was the exclusion rate?

L130-133 The authors switch in their explanations from correct amount order in time to percentiles that can be applied to distribution functions for the correct estimation of precipitation amounts. This requires the assumption that the unknown bias mentioned before does not change over time and between events. This should be communicated to the reader as well.

L159-160 Does this approach introduce an upper limit for the point of interest, resulting from the maximum rainfall amount measured at the surrounding primary stations? Or are theoretical distribution functions applied and the information is missing (or I missed it)?

L228 The first filtering? I would have assumed that this extra-ordinary event-based deviation would have been eliminated by the second filtering.

Fig. 3 & Table 3 From Fig. 3 it is obvious that the minimum resolution is 0.01mm for the Pluvio, while it is 0.1mm for the PWS. This makes a comparison of p0 without its consideration biased. Was the different measurement resolution taken into account for the values of p0 in Table 1? Otherwise I would recommend to either neglect values <0.1mm or to sum rainfall amounts up to a minimum of 0.1mm. The pluvio will gain more dry time steps by doing so. It maybe has a neglectable effect for hourly time steps, but for the original temporal resolution of 5min it will be critical. Hence, it should be at least communicated to the reader.

Fig.3 I recommend to add x-y-lines to illustrate the perfect match since in the left figure it is not the diagonal.

Fig. 5 Indicator correlations with values below the minimum resulting from the primary network for similar distances are included in the right figure. From my understanding these were removed by (2)? Also, for the decision of keeping secondary stations or not indicator correlations for unknown distances resulting from the primary network have to be estimated. In general, this is done by fitting regression lines to the observations? Was it done similar in this study? If so, it could be useful for the reader to provide the type of regression line and it parameters. If not, how were values judged for unknown distances?

L289 "…With increasing…as the role of the bias increases". Is the bias the only reason therefore? I guess the much higher spatial correlation for longer time steps also gives less possibility for improvements, so a frontal event with 12h duration covers some of the stations from the primary network, while this is not the case for hourly time steps (it is mentioned later, L298).

L335-336 Do the authors mean "event" here instead of "data"? Otherwise I'm wondering to not find the values for the RMSE in Table 5.

Technical corrections:

L45 Bracket before the authors.

L62 If the authors refer to the Black Forest, it should be indicated for the reader who is not familiar with the German topography in Fig. 1 where the Black Forest is.

L67 Maybe at least the location of Reutlingen can be shown in Fig. 1

L75 Can the authors provide a reference for the 5°C threshold or how was it chosen?

L80-84 Again, it would be helpful for the reader if mentioned locations are shown in Fig. 1. Otherwise, the locations can be left out.

L94 „This relationship" Which relationship?

L96 „it" is missing

L98 „be be"

L99 „Δt durations" -> „durations Δt"

L100 „Δt" was introduced before.

L105 Maybe „exceedance probability" instead of „probability".

Eq. 1 „Δt" is missing as index on the left side of the equation.

L185 There is no subsection 3.4.2 .

L186 missing comma

Table 1 The criteria/abbreviations should be introduced in the table caption.

L213 we -> were

L218 on -> one

L233 over -> overestimate

Table 3, 4 and 5: Line 2 is redundant with line 2 from Table 2.

L284 Something is missing in this sentence, please double-check.

L349 based -> is based

Author contributions: sduy -> study

---

## Referee Comment (RC5) · Marc Schleiss (Referee) · 10 Mar 2020

Summary: The authors investigate the usefulness of personal weather stations (PWS) for quantitative precipitation estimation, with an emphasis on high-intensity rainfall events. A two-step procedure is proposed in which the first step is to select "good" PWS stations based on spatial consistency tests and the second step is to predict the rainfall rate at a target location using interpolation. The method is applied to a large dataset of professional and PWS data in Germany and the performance is evaluated using cross-validation.

General assessment: The topic is very relevant and the authors have some good ideas

for how to tackle the quality control and interpolation aspect of uncertain measurements from PWS. The 4 referees before me have already done a good job at pointing out relevant issues that need to be fixed/clarified before publication. Here are some additional ones:

Major comments:

a) The authors should provide more details about the kriging part. - How did you estimate the variograms? (with/without zeros?) - How do the variograms look like? - Which variogram model did you use and how well does it fit the empirical variogram? - How do you deal with cases in which there are not enough data to reliably fit a variogram? - How do you deal with spatial anisotropy and intermittency during interpolation?

b) Ordinary kriging makes rather strong assumptions about the data (such as second-order stationarity). The latter might not be very realistic in heavy localized rain events. Kriging is also relatively slow compared with other deterministic interpolation methods and its accuracy strongly depends on the density and number of primary observations. For example, the estimation and fitting of a variogram model (from a small number of samples) might introduce additional errors into your predictions that are due to modeling choices rather than the quality of the data. So my question is: why did you choose ordinary kriging? Please motivate this choice by some form of cost/benefit analysis, for example by comparing it to simpler, faster alternatives such as inverse weighted distance interpolation or bilinear interpolation (which make different modeling assumptions).

c) Related to the previous comment. Please note that during cross-validation, one part of the error is due to the spatial interpolation method that you use (i.e., kriging). If you had taken a different interpolation method (say IDW or Bilinear), perhaps the usefulness of the PWS data would have been different. I think it is important that you assess this part of the error by using at least one alternative non-parametric interpolation method other than kriging (e.g., bilinear interpolation). My point here is that in
some cases, you might see improvement for one particular interpolation method but not for another.

d) The cross-validation part lacks crucial details about parameter estimation. For example, did you use the same variograms or recalculate them based on the selected subset of observations? Theoretically, you should recalculate the variograms on the smaller subset.

e) The second step (i.e., amount estimation) involves a quantile mapping. According to your Figure A1, this mapping is different for each PWS. However, this would mean that you need to estimate and fit a separate variogram model (with different nugget/range/sill) for each PWS location at which you want to interpolate. Is that correct? This would be computationally heavy. Please add more details to help me understand this.

f) Wind is known to cause localized biases in rain gauge measurements in the order of 10-30%. The latter are not stationary over time and space and can significantly affect the ordering of your data, therefore violating your model assumptions (i.e., monotonic link between quantiles of primary and secondary variables). This is not catastrophic but will occasionally affect the accuracy of your rainfall estimates and lower the reliability of your method. I think this issue should be clearly mentioned and discussed in the paper, along with the other limitations in the methodology mentioned by the other reviewers.

g) Tables 3 and 4: Your evaluation of the improvement in terms of a binary response (yes/no) is not very informative. Improved by how much? Some conditional error distributions (for both cases) might help shed some more light on best/worst case scenarios and what to expect in practice.

h) I agree with Lotte de Vos (referee 1) when she says that more details about the limitations of the method need to provided. I would go one step further and say that right now, the paper is heavily focused (biased?) towards demonstrating potential and improvements over the status quo. However, the numbers suggest there are also a lot

of cases in which the PWS data deteriorate the accuracy of the predictions. Perhaps you could show a few of these cases and comment on them. By explicitly showing what can go wrong, you may be able to provide concrete recommendations for future developments.

Minor comments:

1) Details about how correlation is calculated from binary variables are missing. I assume you used the Pearson linear correlation coefficient. But other choices are possible. Please clarify.

2) If a PWS gets accepted for a high value of alpha but not for a lower one, what does this mean?

3) Page 10, l.236: [. . .] indicating that the occurrence of precipitation can be well detected by the secondary network. This statement is misleading. According to previous studies, one of the most common type of errors in PWS are faulty zeros (i.e., the PWS gives zero rainfall during rainy conditions). The latter can occur without warning and only last for a few time steps (e.g., due to data transmission problems or temporary shielding of the gauge). The PWSs that pass the QC tests can still contain faulty zeros. Please reformulate to convey the right meaning.

4) Figure 5: How do you explain the red crosses that have higher correlation than the black crosses? Especially the ones with correlation coefficients close to 1. Are these computed from very small samples?

5) Appendix A, page 20, line 397: The equal weights of $\frac{1}{4}$ only applies to the case of an isotropic variogram.

---

## Author Comment (AC1) · 26 Mar 2020

We thank Hannes Müller-Thomy for his thoughtful remarks. Our answers to the specific comments (in blue) are as follows:

L26-28 The short periods of available radar data should be mentioned in this context as well.

We will add this aspect in the revised manuscript.

L77-79 It would be helpful if the authors are more concise regarding the number os

[Figure]

PWS stations finally used in the study. To enable a transfer of the applied methods the authors should provide some information, which minimum time series length was chosen for the secondary time series and how was it chosen?

The number of PWS stations varies strongly due to the increase of the network in time and due to unexpected missing records. The first filter is used to identify the locations which can be used. The number of PWS for each time step is normally slightly less than the number of stations remaining after the first filter. This depends on which stations had valid observations for this time step and if they were eliminated by the on-event filter or not. We'll include the actual number of PWS used for the maps presented in the paper. The minimum length of observations for the application of the first filter was two months. This is a reasonable choice for hourly aggregations. For longer aggregations longer series would be required. This of course leads to high uncertainties of the indicator correlations.

L85 From the first paragraph in Section 3 it sounds as only the two data quality filters will be explained. I suggest to provide a brief overview of all subsections at the beginning of Section 3 and an explanation, how they are interacting.

Referee 2 suggested a flow chart to illustrate the procedure, we will make this more clear at the beginning of the Methodology chapter.

L102 Maybe the authors should explain briefly why they consider Y as a random field.

This is a mistake and will be corrected, Y is not a stationary random field. It is the sum of precipitation (considered as random field) and a measurement error which is spatially independent, temporally dependent and has a non-zero mean.

L120-123 The chosen criterion sounds reasonable. I'm wondering if an exclusion for too high correlations has to be applied as well. Later in Fig. 5 indicator correlations of 1 are shown for interstation distances of 10 km, which is way higher than from the primary network. Maybe the authors can report if an upper limit is required or not when working with the data as a result from their data analysis.

Due to the partly very short time series the indicator correlation between the primary and secondary networks can fluctuate a lot. We did not calculate the sample size dependent confidence intervals of the correlations as this should be done for each pair individually. Instead we decided to remove the low ones - where we certainly removed a few which provide reasonable data. The correlation being 1 is mainly the consequence of small samples, and thus we did not exclude those stations.

Also, I'm struggling with the final decision if a secondary time series remains in the potential useful data set or not. As far as I understand it a time series is "flagged as suspicious" if it does not meet the criterion in Eq. 2. That means the time series will be sorted out. Since the procedure is repeated for several $\alpha$ and $\Delta t$, I imagine the highest exclusion rate will be found for high values of $\alpha$. Is a flagging for only one of the analysed values of $\alpha$ enough for an exclusion of that time series? Which values of $\alpha$ have been applied and what was the exclusion rate?

Due to sample size we decided to apply the filter to the hourly data. The reason for taking the 99 % threshold was that we are mainly interested in heavy rainfall. Other durations and thresholds were also calculated but the decision was taken on the basis of the above aggregation and threshold. For these, the exclusion rate was about 60%.

L159-160 Does this approach introduce an upper limit for the point of interest, resulting from the maximum rainfall amount measured at the surrounding primary stations? Or are theoretical distribution functions applied and the information is missing (or I missed

it)?

There is no upper limit on the observations implied by the second filter. If the second filter is applied on the percentiles the upper limit is 1.

Fig. 3 & Table 3 From Fig. 3 it is obvious that the minimum resolution is 0.01mm for the Pluvio, while it is 0.1mm for the PWS. This makes a comparison of p0 without its consideration biased. Was the different measurement resolution taken into account for the values of p0 in Table 1? Otherwise I would recommend to either neglect values <0.1mm or to sum rainfall amounts up to a minimum of 0.1mm. The Pluvio will gain more dry time steps by doing so. It maybe has a negligible effect for hourly time steps, but for the original temporal resolution of 5min it will be critical. Hence, it should be at least communicated to the reader.

Thank you for pointing out the issue with the zeros and the resolution. This effect is indeed critical for high temporal resolution, i.e. 5 Minutes. In Fig. 3, we will consider this aspect by summing up amounts from the Pluvio to 0.1mm. The numbers in table 1 are based on 1h resolution, so this effect should be negligible, but we will check this and correct it if necessary.

Fig.3 I recommend to add x-y-lines to illustrate the perfect match since in the left figure it is not the diagonal.

We will add this.

Fig. 5 Indicator correlations with values below the minimum resulting from the primary network for similar distances are included in the right figure. From my understanding these were removed by (2)? Also, for the decision of keeping secondary stations or not indicator correlations for unknown distances resulting from the primary network have

to be estimated. In general, this is done by fitting regression lines to the observations? Was it done similar in this study? If so, it could be useful for the reader to provide the type of regression line and it parameters. If not, how were values judged for unknown distances?

The indicator correlation are filtered by comparing the correlation between the pairs (1) PWS station - Primary neighbouring station and (2) Primary neighbouring station - Primary neighbouring station. This is done for the available PWS time period and varies individually. This is why, the equation was tested for each PWS and fitting a regression line cannot describe the individual behaviour between each PWS and it's neighbours.

L289 "...With increasing...as the role of the bias increases. Is the bias the only reason therefore? I guess the much higher spatial correlation for longer time steps also gives less possibility for improvements, so a frontal event with 12h duration covers some of the stations from the primary network, while this is not the case for hourly time steps (it is mentioned later, L298).

The aggregation leads to more smooth and higher correlated variables which is as the reviewer pointed out another reason for the smaller improvements for longer aggregations. This will be mentioned in the revised paper.

L335-336 Do the authors mean "event" here instead of "data"? Otherwise I'm wondering to not find the values for the RMSE in Table 5.

Table 5 contains the RMSE calculated over all events and stations, while in the text discussing the figures we used the RMSE calculated for the single event using all available primary stations. That is why the numbers are different. The word data will be replaced by event.

L75 Can the authors provide a reference for the 5°C threshold or how was it chosen?

This threshold was chosen arbitrarily, we wanted to be sure not to include any snow fall events, so this is threshold is rather strict.

The other technical corrections will be considered while preparing the revised manuscript.

---

## Author Comment (AC2) · 26 Mar 2020

We thank the anonymous referee for positive review and for the suggested corrections.

2. Figure 1: Red triangles are difficult see against brown elevations. Please consider changing colour, e.g. to black and bigger triangles

We will revise this figure accordingly, Reviewer 4 also has also recommended some changes

3. Lines 102-103: Why can multivariate methods like Co-Kriging not applied to random

fields?

This sentence will be corrected. The problem for applying Co-Kriging is that co-variograms cannot be calculated in a traditional way as there are no common observation locations between the primary and the secondary networks. We found an interesting reference (Clark et al. 1989) where a non-collocated version of Co-Kriging was presented. We applied this methodology to the filtered data. The results show significant improvements, but the combination of the transformation and the Ordinary Kriging leads to superior results.

4. Lines 146-147ff: The sentence with quantiles and percentiles first caused some confusion to me. After reading several times I understood that the term "quantiles" is used here for precipitation values with certain non-exceedance probabilities (Eq. 5), which is common. But the term "percentiles" is used here for the non-exceedance probabilities (Eq. 4), which is not always common. Often, it also refers to the quantiles which divide the distribution into 100 equal portions. In order to avoid confusion, I would suggest beside giving equation (4) also verbally to make clear that with percentiles the non-exceedance probability is referred to. Please, also make a comment on G(y) and F(x) if here empirical or theoretical distributions will be used.

We will address and clarify these issues in the revised manuscript

5. Equation (5,6): It becomes not immediately clear which x(i) locations are related the y(j) location. Please, explain in the text and make a reference to Appendix A here.

We checked the equations (5,6), the primary observation locations are $x_i$ the secondary $y_j$. This is correct in the equations, but we'll add some text to better explain the procedure.

6. Line 160: The estimate for y at time t can be bigger the observation at this time but cannot be bigger than the maximum observation for all times t at x, if an empirical distribution for F(x) is used. Please comment.

The remark is correct. If one would use fitted theoretical distributions one could obtain *new record* values. The usefulness of this approach has to be tested. We'll add some discussions on this.

7. Line 205: Is there a reference available for KU?

Delhomme (1978), we'll add this reference.

11. Line 277: "There is no improvement..." From Table 3 I see improvement for the different time aggregations between 17% and 60% of the stations?

The 17 % means that in 17 % of the cases the estimation was better and in 83 % of the cases it was worse.

The other minor comments (1., 8., 9.,10.,12. and 13.) will all be considered in the revised version of the manuscript. Furthermore we will add the following references:

Clark, I., Basinger, K. L., and Harper, W. V., 1989, MUCK - a Novel Approach to Co-Kriging,in B. E. Buxton (Ed.), Proc. of the Conf. on Geostatistical, Sensitivity, and Uncertainty Methods for Ground-Water Flow and Radionuclide Transport Modeling: Battelle Press, p. 473–493.

Delhomme, J.: 1978, Kriging in the hydrosciences, Advances in Water Resources, 251–266,
* * *

---

## Author Comment (AC3) · 27 Mar 2020

We thank Nadav Peleg for taking his time to carefully read our paper and for his constructive remarks.

1. The motivation to use PWS in rainfall estimation is quite clear and well written in the introduction. However, many studies suggest various stochastic and deterministic methods to blend/merge/interpolate different rainfall products, e.g. combining data from rain-gauges, weather radar and CML together. Why not applying an already established method to merge data from trustable rain-gauges

with PWS? There should be a short explanation in the introduction of why a new merging method is needed.

Merging data requires assumptions on the dependence of the variables and on the error structure of the secondary variable. In the case of PWS, the errors are spatially independent but due to the fact that most of the measurements are likely to be biased the errors do not have zero as mean. This already reduces the number of possible merging methods. Furthermore, quite a few stations may provide erroneous data; that is why we decided to use a filter first. After filtering, some of the established methods such as Co-Kriging could be applied. We tested a non-collocated version of Co-Kriging and found that the *correction* of the secondary observations leads to better results. We'll address this point in the introduction.

2. Empirical distributions are used for all PWS. I was wondering if it wouldn't be more accurate to use a specific distribution instead. For example, the same distribution can be fitted to all the trustable stations (but with different parameters), and the parameters can be spatially interpolated to the PWS (and other) locations.

This is a good idea and may help identify and to quantify some extremes of the PWS. At the present stage we intended to keep the methodology as simple as possible.

3. I agree that the examples presented in figures 6 to 8 cannot be evaluated against "true-rainfall" due to a lack of spatial information. That is why I believe that there is an added value in comparing the outcomes of the interpolation with data emerging from the weather radar composite in Germany. If you do not trust the radar QPE, there is no need to compare the actual rainfall intensities, but just to demonstrate that the interpolated rainfall fields can assist in revealing high-intensity rainfall features that are "hidden" when using the official rain-gauge network alone.

We compared interpolated rainfall maps with radar images and discovered quite a few cases where the primary network missed intense precipitation which was

detected using the PWS and also appeared on the radar image. We'll add an example for this to the paper.

4. The potential to use PWS to generate rainfall fields at a minutes-scale is very appealing, especially for applications in urban hydrology. I see the potential in using PWS to simulate rainfall fields at high temporal-resolution, but in the presented study no sub- hourly examples are presented. It will be nice to see if the potential to interpolate the rainfall at high-resolution can be fulfilled and to discuss the limitations of the PWS and methods in going to such fine scales.

We did not include any examples for short time scales (5 to 30 minutes). The reason for this is that for very fine time scales a space time interpolation is likely to perform much better than the pure spatial interpolation. This however requires some new theoretical developments including advection direction and speed estimations which go beyond the scope of the present paper.

Minor points

L64. 10-min? Yes, some even 1 Minute. In our study, we aggregated all data (i.e. DWD and PWS) to 1h temporal resolution. We will describe our data processing more thoroughly in the revised manuscript.

Figure 2. It can be presented as Supplementary Material.

Based on the comments from referee 1, we will add an additional sub-figure showing a histogram of the available length of the PWS time series and would therefore like to keep this figure in the main text.

L98. "at short time steps" - 1-min? 5-min?

The PWS data are available at 5-min resolution, c.f. answer to L 64.

L102-103. "...thus methods like Co-Kriging or Kriging with an external drift are not applicable" - at this point in the text, some further explanation is needed to put this sentence in context.

Co-Kriging in its regular form cannot be applied but we found a method to use non-collocated observations which we applied. We'll add a few remarks on Co-Kriging.

L102. "is considered to be a random field" - Why? Reading further, this sentence is clear. But it is not clear at first reading.

It should be "not a stationary random field". Will be corrected.

L105. It should be mentioned in the text that alpha defines the percentile threshold. I assume it is subjectively defined?

This was also remarked by referee 1, we'll address this appropriately in the revisions.

Equation 1. I assume Fu stands for distribution function? Please clarify in the text. In addition, there are two commas with empty space in the left term of the equation.

We will clarify this. In Eq. 2 there's a $\Delta t$ missing between the commas.

Section 3.2. Consider adding a flow chart to illustrate the steps described in this section.

Referee 4 also made a remark that the work flow and interaction of the steps should be pointed out more clearly. We will consider adding a flow chart to make this more clear.

L239. Isn't 95 percentile too low threshold if the goal is to attract the extreme rainfall intensities? Especially for the fine temporal resolution, for which I assume the sample size is quite large.

We tested this for different threshold starting from 95. For the study we've used the 99 percentile.

L397-400. Wouldn't it be more accurate to fit a specific distribution to each secondary station, based on parameters obtained from the primary stations around it?

We do not fit the distribution to the secondary observations as they are biased, but we interpolate the distributions from the primary stations.

The other minor remarks made by the referee will be considered in the revised manuscript.

---

## Author Comment (AC4) · 27 Mar 2020

We thank Lotte de Vos for taking the time to review our manuscript thoroughly. Regarding the summary, we' like to clarify that we investigated 955 individual events (about 200 for each duration), not only 200.

Our response to the major comments:

P2L42-49 ff.

We apologize for the misinterpretation of the paper of de Vos et al. (2019). After careful rereading we recognized that our interpretation was wrong, and we'll correct

the corresponding paragraphs in the revised version of the paper. The filtering is in fact not requiring the actual radar product. On the other hand the bias correction filter SO requires the radar product for the previous time period. This is itself is subject of errors. Further please note that the validation of the precipitation amounts is done on the basis of the radar product, for which the uncertainty and inaccuracy plays an important role. The SO filter provides a kind of regional bias correction, our transformation is correcting each station individually as we have observed that even within a small region significant positive and negative biases may occur. The filters FZ and HI are very similar to our second event based filters. The first filter requires at least a few months of observations - this is a disadvantage, but on the other hand it provides an overall judgement of the individual PWS. As the second filter is applied for each event to all stations which passed the first filter. Thus there is little risk that occasionally bad measurement are not rejected. Our filter is in fact rather strict (conservative) as we remove many stations. We need further work to find the best selection of useful PWS and for the bias correction.

The proposed method is interesting and promising, however there are some significant limitations due to the assumptions in the filters. It can be considered contradictory that the main perceived issue with the QC in previous work (mistakenly) is its dependence on another data source, while this methodology relies on the availability of another data source itself. The PWS are used as an addition to a high quality primary rain gauge network with long observation series in the study area of interest, measuring in high temporal resolution. Such a network may not be readily available everywhere, and this should be mentioned in the discussion more broadly than it is now.

It is true that high quality primary measurements might not be available everywhere. We are testing the methodology on smaller primary datasets to quantify the usefulness of the PWS network.

The paper is very limited in describing how the data is gathered from the Netatmo rain gauges, which measure approximately every 5 minutes. The unprocessed time series that can be collected with the Netatmo API do typically not have fixed time steps and can contain large data gaps. The paper is not clear on how these raw time series are processed into structured aggregated time series at 1, 3, 6, 12 and 24 hour time steps, but does mention in the evaluation of Netatmo data from the experimental set- up with a Pluvio sensor an error resulting from station connectivity. This error is difficult to understand without knowing the process that the authors have used.

We will describe the data used and the processing more clearly in the revised manuscript. The data we downloaded using the Netatmo API did have regular 5-min timesteps, however these we're not always continuous. Such gaps in the data were filled with NaNs. These data were then aggregated to 1h sums and by keeping the NaNs, i.e. any 1h-aggregation with NaNs in-between was considered as NaN. We compared the frequencies of the zero observations of the primary and secondary network and did not find significant differences. This means that the problem of providing 0-s for nan-s was negligible in our case (but we did find occasional occurrences of false zeroes when comparing the 3 Netatmos with the reference at our weather station). Moreover, since each PWS station was verified individually, the missing data were always taken into account and the correspond- ing data from the primary network were considered. All analyses in the study are based on hourly precipitation sums, and all other aggregations were based upon these.

Minor comments:

P4L77: "one can see that many stations have less than one year of observations" - how does that follow (from figure 2 or elsewhere), and why is the proposed methodology not able to accommodate these stations?

We will clarify this in the revisions by adding a figure showing a histogram of the time

lengths of the PWS stations. Furthermore, a certain time length (2 months excluding the winter months) is required for the filters to work.

Section 2 would benefit from more quantitative descriptions of the measurement uncertainty of the sensors that are mentioned, e.g. from technical documentation of these sensors from the supplier.

We will address this aspect in the revisions.

P5L103: "Note that Y is considered to be a random field, and thus methods like Co-Kriging or Kriging with an external drift are not applicable." the purpose of this statement in this context is not entirely clear to me.

Correctly: $Y$ is not a stationary random field as the measurement bias and uncertainty can differ from one station to the other. Meanwhile we found a way to use Co-Kriging - after using a transformation. The reference for non-collocated Kriging is in our response to Reviewer #3. The manuscript will be modified accordingly.

Section 3.1 describes that a secondary station is flagged as suspicious if its indicator correlations with the nearest primary network points are below the lowest indicator correlation corresponding to the primary network for the same time steps and at the same separation distance. I can imagine that not all distances between secondary station and nearest primary network points equal a separation distance between two primary network stations exactly. Is then the nearest distance used? If so, what are the largest differences between separation distances? Or is the relationship between distance and correlation ($\rho$) described with a fitted relation (effectively a correlogram)? If so, what is then the meaning of "min" in Eq. (2)?

Each secondary station has a single closest primary station. The indicator correlations are calculated based on the whole time series (after removal of the NaNs) of these

pairs . The indicator correlations using all pairs of primary stations are also calculated using exactly the same timesteps. We assume that the indicator correlations of the primary stations represent the *true* spatial variability of precipitation. Thus we compare these clouds and reject all secondary stations where the correlations are below those primary pairs within a distance window with a tolerance. The tolerance is needed for close pairs of primary and secondary stations. We do not calculate indicator correlations for pairs of secondary stations.

P7L163: ".. due to unforeseen events (such as battery failure or transmission errors) at certain times they may deliver individual false values." → How is the issue of data gaps in Netatmo time series addressed? Here it seems to be referred to as "false values", however it should be evident from the Netatmo time series that an observation was lacking (due to a long duration between the timestamps of two subsequent observations). I wonder if regarding these observations as zero observations and subsequently identifying them with a simple geostatistical outlier detection method is the best approach. The author's may refer to the station in total(not a certain period in observations), which due to battery failure or transmission errors is considered to be faulty. If that is the case, which fraction of the data should be missing for a station to be considered a geostatistical outlier?A later section (P9L224-229) hints at problems due to data gaps which resulted in a large outlier, but it's not clear if these cannot be avoided by looking at the timestamps of the PWS observations. More information on how the raw irregular Netatmo PWS datasets are converted to timeseries with fixed timesteps would be very helpful.

As mentioned above, all missing time stamps in the downloaded data were flagged as NaN (not 0). The timesteps from the data we downloaded are in regular 5-min intervals. We will describe our data processing more clearly in the revised manuscript. In table 1, only 1h timesteps where all devices (i.e. the three Netatmo and the Pluvio reference) have valid data were considered.

Figure 4: why are the lines of the Secondary Stations stepped and the Primary Stations not?

Because of the different resolution of the rain gauges, i.e. Netamo 0.1mm and Pluvio 0.01mm

Table 2 caption: I assume that p0 still refers to probability of precipitation. Is it then the fraction of intervals where precipitation is larger than 0.1 mm? In that case it makes more sense to change the text in the table from "<0.1 mm" to ">0.1 mm". Also, "(mean of all stations and events)" is not very clear in this context, please explain.

P12L260: "Note the high portion of zeros" - where can this portion be found? It doesn't seem to be provided in Table 2. Should this be portions of intervals where precipitation is <0.1 mm?

We will clarify this in the revision.

Table 2: what was the procedure to select these events?

The intense rainfall events were selected from the observation of the primary network. For each temporal aggregation, we investigated the highest 200 intense events. These were selected regardless of the observed location or time. For the cross validation procedure, only events without nugget variograms were chosen, this is why for each temporal resolution the final number of events was slightly less than 200.

P12L274: "Pearson (r) and Spearman ($\rho$) correlation→ up until now I would have assumed the correlation that was introduced in section 3.1 to be the Pearson correlation. However, as the symbol $\rho$ was used in that section, that was likely actually Spearman. Either way, it should be specified in section 3.1. Also, what is the motivation to evaluate

two types of correlation?

As the distribution of precipitation amounts is skewed the Pearson correlation may be strongly influenced by a few high values. The Spearman correlation is independent of the distribution and shows whether the ranks of the observations were correctly reproduced. As our method is strongly based on rank based assumptions it is reasonable to consider it. The text will be revised to recognize which correlation was actually used.

Section 4.2: It is explained that two references are constructed using cross validation. Reference 1 is constructed by interpolating the subsets with only primary network stations, and Reference 2 is constructed by interpolating the subsets with primary and secondary network stations. What is the reason for constructing two references? From their captions it seems that Table 3 and 4 are based on comparisons with Reference 1. Is Reference 2 used somewhere else?

This seems to be a misunderstanding. These sets are not references these are the interpolations - we used a cross validation approach and both interpolations are compared on the observed primary dataset (every time for the stations not considered).

P17L334: "This is caused by the reduction of the variability with increasing number of observations" → Is that true? Why would the variability of a rainfall event be dictated by the number of observations in space? It seems to refer to the more smooth rainfall patterns found at daily scales compared to hourly scales, but this phrasing is confusing.

Our wording is in fact confusing - we meant with the increase of aggregation (the number of 5 min data considered) the fields become smoother. We'll correct this in the manuscript.

[Figure]

P20L400: "The precipitation quantiles at the primary stations corresponding to the 0.99 probability are 3.2, 3.5, 3.1 and 3.0 mm." → how does this follow from the information that is provided? Or is this provided information?

The quantiles are derived from the distributions based on the time series of the primary stations. For this example we assumed that these are the corresponding values.

Some interesting additional literature to refer to could be: https://www.nat-hazards-earth-syst-sci.net/20/299/2020/nhess-20-299-2020.pdf on the use of Ne-tatmo data for describing deep convection features. Also, the QC method https://github.com/metno/TITAN could be mentioned in addition to the QC method of Napoly et al. in the introduction. Finally, Chen et al. (2018) "Trust me, my neighbors say it's raining outside: Ensuring data trustworthiness for crowdsourced weather stations." is an example for quality estimation of PWS rainfall data from the Wundermap platform.

Thank you for pointing out these references, we will consider them in the Introduction.

The other minor remarks will be considered while preparing the revised manuscript.
* * *

---

## Author Comment (AC5) · 2 Apr 2020

We thank Marc Schleiss for taking his time to carefully read our paper and for his interesting discussion on the methodology. Here are our responses to the major comments

a) The authors should provide more details about the kriging part. - How did you estimate the variograms? (with/without zeros?) - How do the variograms look like? - Which variogram model did you use and how well does it fit the empirical variogram? - How do you deal with cases in which there are not enough data to reliably fit a variogram? - How do you deal with spatial anisotropy and intermittency during interpolation?

The variograms used for this paper were calculated using the observations of the primary network only. The variograms were calculated on in the rank space which leads more robust results (Lebrenz and Bárdossy 2017). Further as the kriging weights do not change if the variogram is multiplied by a constant in this study the estimation of the range of the variogram was the major task. We assumed that there is no nugget (precipitation amounts are spatially continuous). The possible measurement error was included in the kriging with uncertainty. Anisotropy was not considered, the main reason for this was that the primary network did not give robust results. In the future we intend to estimate anisotropy from the corresponding radar images. The kriging weights are not very sensitive to the choice of the range and the variogram type as it was investigated in the paper (Bárdossy 1988). The variograms used for the second filter are the rescaled (adjusted to the variance of the observed event) variograms calculated from the percentiles. A discussion on the variogram calculation and fitting including the corresponding references will be added to the paper.

**b)** Ordinary kriging makes rather strong assumptions about the data (such as second-order stationarity). The latter might not be very realistic in heavy localized rain events. Kriging is also relatively slow compared with other deterministic interpolation methods and its accuracy strongly depends on the density and number of primary observations. For example, the estimation and fitting of a variogram model (from a small number of samples) might introduce additional errors into your predictions that are due to modeling choices rather than the quality of the data. So my question is: why did you choose ordinary kriging? Please motivate this choice by some form of cost/benefit analysis, for example by comparing it to simpler, faster alternatives such as inverse weighted distance interpolation or bilinear interpolation (which make different modeling assumptions).

**c)** Related to the previous comment. Please note that during cross-validation, one part of the error is due to the spatial interpolation method that you use (i.e., kriging). If you had taken a different interpolation method (say IDW or Bilinear), perhaps the usefulness of the PWS data would have been different. I think it is important that you assess this part of the error by using at least one alternative non-parametric interpolation method other than kriging (e.g., bilinear interpolation). My point here is that in some cases, you might see improvement for one particular interpolation method but not for another.

Regarding both comments above, we assume local second order stationarity - this means kriging is carried out using a few neighbouring stations only. The assumption partly accounts for the non-stationarity. There are several studies which compared different interpolation methods for precipitation which in most cases showed that kriging is superior to other techniques. We compared the interpolation with inverse distance and nearest neighbour for the selected events. For all three interpolation methods the usage of the filtered and corrected PWS lead to an improvement of the cross validation. The selected OK approach was superior to the others. We did not want to overload the paper with the other interpolation results. We also tested different Co-Kriging approaches which also lead to improvements, compared to the inverse distance and nearest neighbour interpolations, but remains slightly inferior to the simplest OK approach. Therefore not to overload the paper these results are not included.

**d)** The cross-validation part lacks crucial details about parameter estimation. For example, did you use the same variograms or recalculate them based on the selected subset of observations? Theoretically, you should recalculate the variograms on the smaller subset.

Variograms were recalculated for each subset. Due to the relatively large number of primary stations and the fact that we used percentiles the change in

the ranges was minor.

**e)** The second step (i.e., amount estimation) involves a quantile mapping. Accord- ing to your Figure A1, this mapping is different for each PWS. However, this would mean that you need to estimate and fit a separate variogram model (with different nugget/range/sill) for each PWS location at which you want to interpolate. Is that correct? This would be computationally heavy. Please add more details to help me understand this.

Variograms of the quantiles are estimated from the primary stations only. Thus there is no need to recalculate the variograms for each PWS. The appropriate quantiles are also estimated from the primary stations for each PWS locations. For each event this requires one additional OK. The example in the Appendix shows the procedure.

**f)** Wind is known to cause localized biases in rain gauge measurements in the order of 10-30%. The latter are not stationary over time and space and can significantly affect the ordering of your data, therefore violating your model assumptions (i.e., monotonic link between quantiles of primary and secondary variables). This is not catastrophic but will occasionally affect the accuracy of your rainfall estimates and lower the reliability of your method. I think this issue should be clearly mentioned and discussed in the paper, along with the other limitations in the methodology mentioned by the other reviewers.

You are right - wind has a strong effect on precipitation bias. However this applies for both networks. Our methodology is presently focussing on adjusting the PWS to the primary network. We intend to consider wind dependent corrections in the future. Several PWS measure local wind speed this could be used for further investigations.

**g)** Tables 3 and 4: Your evaluation of the improvement in terms of a binary response (yes/no) is not very informative. Improved by how much? Some conditional error distributions (for both cases) might help shed some more light on best/worst case scenarios and what to expect in practice.

We'll add one or two figures on showing error distributions. The main reason for this is to provide a transparent evaluation showing that for the majority of the stations and events there is an improvement, but not for all.

**h)** I agree with Lotte de Vos (referee 1) when she says that more details about the limitations of the method need to provided. I would go one step further and say that right now, the paper is heavily focused (biased?) towards demonstrating potential and improvements over the status quo. However, the numbers suggest there are also a lot of cases in which the PWS data deteriorate the accuracy of the predictions. Perhaps you could show a few of these cases and comment on them. By explicitly showing what can go wrong, you may be able to provide concrete recommendations for future developments.

We do not agree that the paper would be optimistically biased. In Tables 3 and 4 (which you previously criticized) we show the frequencies of cases when the method was better and when it was worse than the standard. This information is usually not provided and shows that there are cases and locations where there are no improvements. Summary statistics as in Table 5 are usually shown and do not provide this information. The locations with no improvements can easily be identified as those where the density of PWS is small. The reason why the PWS bring no improvements for some events is not clear. As the these cases are rare ($< 10$ % for short durations) we do not consider this as a major drawback. Of course further research is needed to improve the interpolation, but we believe that the current results are encouraging.

The minor comments will be considered while preparing the revised manuscript.

Bárdossy,A., Notes on the robustness of the kriging system, *Mathematical Geology,* Vol. 20, No.3, pp 189-203, 1988

Lebrenz, H. and A. Bárdossy, Estimation of the variogram using Kendall's tau for a robust geostatistical interpolation, *Journal of Hydrologic Engineering*, **22**, 2017

---

## Referee Report (RR1)

**"The use of personal weather station observation for improving precipitation estimation and interpolation"**

by András Bárdossy, Jochen Seidel and Abbas El Hachem

Review by Dr. Marc Schleiss, Dept. of Geoscience and Remote Sensing, Delft University of Technology, the Netherlands

**General assessment:**
This is the second time that I review this paper. As I said, I find this topic very interesting and relevant. The large number and variety of available PWS data has created lots of new opportunities and generated a strong need for new, robust interpolation/merging methods. The authors have some good ideas for how to approach the problem. However, I don't think that their study is ready to be published yet. The main points that need to be improved are 1) the writing, 2) the presentation of the results and 3) the description of the methods. Below, please find a list of suggestions for how to improve.

In addition, I should point out that there are 2 major comments from the previous round of review that were not fully addressed during revision. These are:
a) More details about the kriging part → The authors responded to this comment but not of all their explanations can be found in the revised paper. Please make sure that all important details are in the text so that others can reproduce what you did!
b) Comparison of kriging with simpler, faster alternatives such as inverse distance weighted interpolation or bilinear interpolation → Partially done but results are not shown and there's only a few short sentences in the paper about this, without any numbers or critical discussion about the pros/cons.

**Recommendation:** Major Review

**Major comments:**
Note that all referenced page/line numbers refer to the revised manuscript with track changes.

1) Please clearly state the main main conclusion of your paper in the abstract and conclusions. Right now, this is not 100% clear. Is the conclusion that careful QC and bias-correction has to be performed before PWS precipitation data can be used? If that's the case, then this is not really new. Other studies have already shown the same and your method is just another way to do this. So what exactly is your contribution? Please clarify!

2) Your method is rather complicated. Yet several of its components do not seem to significantly improve performance. For example, the EBF filters and the KU do not make a big difference. So why did you feel the need to include them in the methods and results? It just makes the paper longer and more complicated and forces you to introduce a lot of theory and notations for no obvious gain in performance. I suggest to shorten the paper and only keep the essential parts of the algorithm in the methods section. If you want, you can always write a short section or paragraph summarizing the results for some other options/filters that you think could be useful in other contexts.

3) The number of peer-reviewed studies about PWS and their use in hydrometeoroloy is still limited. A few of them have already been mentioned in the literature review. But overall, the introduction of the paper remains rather short. I suggest to extend this part by providing a more in-depth analysis and discussion of the state-of-the-art related to the use of citizen gauges in quantitative precipitation

estimation problems, including its challenges, similarities with other fields and open questions. For example, some parts of the Discussion (i.e., the differences/similarities with radar-gauge QPE) could be moved to the introduction. Also, I encourage the authors to explicitly state which aspect(s) of the problem their study is meant to address. What's the main contribution? Is it the method itself or is it the lessons learned and/or recommendations for a successful interpolation/merging of PWS data?

Suggested references (non-exhaustive):

- Canli, E., Loigge, B. & Glade, T. Spatially distributed rainfall information and its potential for regional landslide early warning systems. *Nat Hazards* **91,** 103-127 (2018). https://doi.org/10.1007/s11069-017-2953-9

- Reges, H. W., Doesken, N., Turner, J., Newman, N., Bergantino, A. and Schwalbe, Z. CoCoRaHS: The Evolution and Accomplishments of a Volunteer Rain Gauge Network, *Bull. Amer. Meteor. Soc.* (2016) **97** (10): 1831–1846. https://doi.org/10.1175/BAMS-D-14-00213.1

- Simpson, MJ, Hirsch, A, Grempler, K, Lupo, A. The importance of choosing precipitation datasets. *Hydrological Processes*. 31: 4600-4612 (2017). https://doi.org/10.1002/hyp.11381

- Starkey, E., Parkin, G., Birkinshaw, S., Large, A., Quinn, P. and Gibson, C. Demonstrating the value of community-based ('citizen science') observations for catchment modelling and characterisation, *Journal of Hydrology*, 548, 801-817 (2017). https://doi.org/10.1016/j.jhydrol.2017.03.019.

- Yang, P., & Ng, T. L. Fast Bayesian regression kriging method for real-time merging of radar, rain gauge, and crowdsourced rainfall data. *Water Resources Research*, 55, 3194– 3214 (2019). https://doi.org/10.1029/2018WR023857

4) The writing and structure of the Results section need to be improved. The current strategy for assessing/validating the different components of the method is not clear to me. Right now, analyses/results are presented in seemingly random order, with rather vague qualitative statements and lots of circumstantial evidence. A better, more precise, quantitative and targeted evaluation would greatly increase the quality of the paper. For example, you could consider a step-by-step, hierarchical assessment of the different components (e.g., the IBF filter, the bias correction and the interpolation/merging), with different scores and subsections for each part.

5) Figure A1 is crucial for understanding how the bias adjustment method works. I suggest to move this from the Appendix to the main text, together with the corresponding explanations. Actually, I don't think you need an appendix at all!

6) Table 3 does not show correlations (which should be between -1 and 1). Please correct.

7) The step-by-step description of the algorithm is a good idea. But it's really hard to follow, even for somebody familiar with the geostatistical jargon. More work is needed to streamline this and make it clear. A flowchart of the whole method would help, with different symbols for filters, adjustments and interpolations! Also, you could shorten the text by grouping some of the smaller steps together into larger modules or tasks. The details of each task can be given in the different subsections of the methodology.

8) The crucial assumption behind your method is that for high precipitation intensities, the ranks of the secondary stations are correct. Some superficial analyses in Section 4.1 suggest that this assumption is probably not too bad. But since this is such a critical hypothesis, it should be assessed in much more detail. Please extend Section 4.1 and perform more tests designed to assess how good this ordering assumption really is. For example, your could compute rank correlation coefficients for different thresholds, stations and lengths of time series. Or you could look at fluctuations over time or as a function of distance. To better understand the limitations of your method, it could also be good to show a few cases for which the assumption does not hold.

9) I have some issues with the terminology chosen by the authors, especially regarding the EBF (Event-based filter). I think this is a poor choice of words. In reality, the EBF filter is a spatial filter for one particular aggregation time period (and not an event). More generally, I don't think that it is a good idea to use the word "event" to refer to a particular aggregation time periods. This is not standard practice and might be confusing to many readers. Please modify accordingly.

10) Regarding the bias correction scheme: If I understood the approach correctly, the idea is to use the percentile of the PWS observations (secondary network) to estimate the equivalent precipitation estimates of the professional gauges (primary network) and then spatially interpolate this value to the location of the PWS using kriging. On top of the large uncertainty that comes with estimating a percentile from a short PWS series, one problem with this approach is that it uses the ordering assumption multiple times (i.e., once for each pair or PWS and professional gauge). This greatly increases the chances of errors during bias correction due to imperfect modeling assumptions. Also, the final spatial interpolation may re-introduce bias due to smoothing and/or modeling choices. So my question is: why don't you just pool the professional rain gauge data together into a single distribution and directly adjust the PWS observations using quantile-quantile mapping on the pooled data? In this way, you would use the ordering assumption only once and you would not have to interpolate at all, which is likely to be faster and more robust. By the way, you can pool data even if the time series of the professional gauges have different lengths. Please explain why you think the current approach is better!

11) A substantial part of Section 5 (Discussion) from lines 434-455 is not a discussion but just a summary of the method and therefore should be moved to the conclusions. The last part of the discussion (ll.467-475) about the similarities/differences of PWS with radar measurements. This is out of scope here because not part of the analyses. I suggest to shorten this and/or move it to the introduction. Please use the discussion section to analyze pros/cons, mention alternatives or new ideas for follow-up studies.

12) Conclusions, ll.501-503: Wind has a major effect on precipitation measurements, leading to a systematic undercatch. This may influence the order of data, but the effect is the same for the primary and secondary network."

> I do not agree with this statement. Literature shows that wind effects tend to be very local. Sometimes, both gauges will be affected by the same bias. But often, it's likely that the PWS and professional gauges will have different biases. More importantly, wind-induced biases will fluctuate over time and space, which affects the rank statistics and the performance of the IBF and bias correction schemes. There's not much that you can do about this. But at least, you should properly acknowledge the problem and discuss its possible consequences in the text. I suggest to do this in Section 5 (Discussion) rather than the conclusions.

13) On a personal note: PWS stations tend to cluster in/around urban areas. Spatial interpolation methods such as kriging do not always perform optimally on highly clustered data. For example, it is well known that clustering can lead to screening effects and highly negative kriging weights. This does not necessarily lead to wrong estimates but decreases robustness and accuracy. I am aware that this goes beyond the scope of this study. Still, I invite the authors to briefly mention this issue in the Discussion section and to point to possible ways to overcome it in future work. This is particularly relevant for small-scale estimates of heavy precipitation.

**Minor comments and typos:**

- In the abstract, please specify what you mean by secondary observations. I assume it's the observations from the PWSs.

- Introduction: "In recent years, the amount of low-cost personal weather stations (PWS) has increased with an incredible speed".

      Incredible is not a good choice of words here. Please reformulate.

- Introduction, ll.24-26, "This is potentially very useful to complement systematic weather observations of national weather services, especially with respect to precipitation, which is highly variable in space and time".

      Please add a few references at the end of this sentence to support your statement.

- Introduction, ll.28-29, "In consequence, the number of interpolated precipitation products with sub-daily resolution is low, but such data  are required for many hydrological applications (Lewis et al., 2018)"

- Introduction, ll.29-31,"Additional information such as radar measurements can improve interpolation (Haberlandt, 2007), however, radar rainfall estimates are  still highly prone to different kinds of errors (Villarini and Krajewski, 2010) [...]"

- Introduction, ll.33-34, "However, one of the major drawbacks from PWS precipitation data is their trustworthiness"

      Please add a few references to support this statement.

- Introduction, ll.36-37, "The measured data itself may have unknown errors which can be biased and contain independent measurement errors, too."

      This sentence is not clear. Please reformulate and be more specific.

- Introduction, ll.45-47, In a more recent study, de Vos et al. (2019) developed a QC methodology of PWS precipitation measurements based on filters which detect faulty zeroes, high influxes and stations outliers based  on a comparison between neighbouring stations.

- Section 2, l.69, "The gauges used in this network are typically weighing gauges".

Do you mean predominantly? In addition, please specify the type of weighing gauges (e.g., the model, brand or serial number).

- On l.119, you mention that the random variable Y is not stationary. Yet, on ll.144-145 and Equation 2, you refer to its cumulative distribution function F, without any dependence on time. Please clarify this apparent contradiction.

- Equation 4, what's your definition of "nearly" at the same separation? Please specify!

- l.168, "Under the assumption that the temporal order of precipitation at secondary locations is correct"

- ll.168-172, "Under the assumption that the temporal order of precipitation at secondary is correct (eq.1), one could have used rank correlations instead of the indicator correlations. The indicator approach is preferred however, as the sensitivity of the devices of the primary and secondary networks is different and this would influence the order of the small values strongly. Furthermore, random measurement errors would also influence the order of low values. In order to have a sufficient sample size and to have robust results, high α values and low temporal aggregations Δt are preferred."

Or you could just say that the ordering between the primary and secondary networks needs to be the same for values above a certain threshold.

- Section 3.3, ll.215-216 "Instead one can use rank based methods for this purpose as suggested in Lebrenz and Bárdossy (2017) and rescale the rank based variogram "

- Section 3.5, ll.279, "Interpolate precipitation for target grid using all remaining values using OK or KU."

Bad English, please reformulate

- Section 4.2, ll.375-376 "Decreasing spatial variability and increasing regularity with increasing time aggregation is the reason for these differences."

I am not sure to understand what you mean by regularity. Please reformulate to make this clear.

- Section 4.3, ll.395-397 "Note that for this data the cross validation based on the primary observations showed an improvement of r from 0.36 to 0.77, of r S from 0.55 to 0.76 and a reduction of the RMSE from 12.5 to 8.2."

Units for the RMSE values are missing. Same for line 409.

- Section 4.3, ll.398-399 "Figure 7 shows the distributions of the cross validation errors for the different interpolations for this event. This is a typical case where all methods yield unbiased results "

- Section 5, ll.440-442 "This approach uses a comparison of the data with those of the nearby stations to remove unreasonable values, a separate procedure to identify and remove false zeros and another  filter to find unreasonably high values."

- Section 5, ll.452-455 "The use of secondary stations after filtering and data transformation improves the results of interpolation for other possible interpolation methods, such as nearest neighbour or inverse distance weighting. However, in this study these methods yield worse results than OK (results not shown here)."

Not clear. Please provide more details. For example, you could give the average reduction in terms of RMSE or increase in correlation for each interpolation method.

- Figure 1: Please add a scale! Same comment for figures 6, 8, 9,10

- Figure 3: Please use different symbols for N07, N10 and N11 to better distinguish the points.

- Figure 4: Please specify the 3 primary and 4 secondary stations in the caption and how far away they are from each other.

---

## Referee Report (RR2)

**"The use of personal weather station observation for improving precipitation estimation and interpolation"**

by András Bárdossy, Jochen Seidel and Abbas El Hachem

Review by Dr. Marc Schleiss, Dept. of Geoscience and Remote Sensing, Delft University of Technology, the Netherlands

**General assessment:**
This is the third time that I review this paper. The paper has improved. However, there are still a lot of typos and unclear sentences and the writing could be improved further. Most of the major issues I raised during the previous rounds were (partially) addressed. The only major points of criticism that I have left are:
- the justification of the assumptions in Section 4.1, which could be more quantitative and exhaustive.
- the conclusion section, which is too short and does not include all major findings.
- the structure of the paper. In particular, the event selection procedure and cross-validation strategies which should not be in the results part but introduced earlier in the text, in the methods section.

**Minor comments and typos:**

- Conclusions: Your conclusion section is really short and does not do justice to all the work that you have done. I suggest to extend it. For example, you do not mention two crucial points which are: a) the gain in performance when using KU compared with OK and b) the fact that important assumptions where made during the development of the filters, such as stationarity, preservation of the ordering of the data and neglecting of other sources of errors such as local wind effects.

- *"Although this is only one example with a relatively short time period it does support our assumption that the quantiles between primary and secondary stations are similar for higher precipitation intensities. However, one secondary device (N10) delivered data which deviates substantially from the other measurements. This was caused by an interrupted connection between the rain sensor and the base station. In this case, the total sum of precipitation over a longer time period was transferred at once (i.e. in one single measurement interval) when the connection was established again. This leads to an extreme outlier which falsifies the results."*

> Not sure to understand your argument here. According to your assumption, the points in Figure 4b should align with each other (though not necessarily along y=x). Still, there seems to be substantial residual scatter and uncertainty due to quantization effects (especially for Netatmo). Please provide some quantiative metrics to judge the degree of linear relationship and highlight which data point in 4b corresponds to the "extreme outlier". In addition, it would be worth commenting on the discretization effects you see in the Netatmo stations.

- ll.73-74: *"The number of secondary stations is higher in densely populated areas  such as in the Stuttgart metropolitan area and the Rhine-Neckar Metropolitan Region between Karlsruhe and Mannheim."*

- ll.103-104: *"It is assumed that this precipitation is measured by the primary network [..]"*

- ll.105-107: *"The basic assumption for the suggested quality control and bias correction method is that the measured precipitation data from the secondary network may be biased in their values but correct in terms of their order "*

- l.119: "As a first step in quality control, all PWS  with  notoriously inconsistent rainfall values are removed."

- ll.149-150: *"In order to have a sufficient sample size and to have robust results, high α values and low temporal aggregations Δt are preferred."*

    Can you be more specific? What are sufficiently large values for alpha and delta t?

- ll.163-164: *"Another possibility is to interpolate the quantiles corresponding to selected non percentiles or interpolating percentiles for selected precipitation amounts."*

    Not clear. Please reformulate.

- ll.272-273: *"Furthermore, one can observe that the differences  between the reference and the Netatmo gauge are not linear, ..."*

- l.276: *"Figure 4 shows that for high percentiles their occurrence is the same for the primary and the secondary devices."*

    This sentence is not clear. Please reformulate.

- ll.289-291 *"While the distributions differ, the probability of no precipitation p0 (defined as precipitation < 0.1 mm) ranges from 0.90 to 0.91 and is thus very similar for both types of stations indicating that the occurrence of precipitation can be well detected by the secondary network."*

    Actually, in Table 1, the percentages p0 (at 1h resolution) are 0.84 for N07 and N10, which is 7% lower than for N11 (0.91) and 8% lower than for the Pluvio (0.92). Please explain!

- l.302 *"In our case, 862 secondary stations remained after the application of the IBF."*

    In addition to the number, please specify the percentage of stations that were removed.

- ll.316-317 *"The secondary station in the centre recorded 1.7 mm of rainfall"*

- ll.332-334 *"The cross validation was carried out for a set of different temporal aggregations Δt and a set of selected events. Only times with intense precipitation were selected, as for low-intensity cases the interpolation based on the primary network is sufficiently accurate"*

    Actually, you did not show any evidence that the interpolation for lower intensities is accurate. Please provide some numbers or reformulate this sentence.

- ll.338-348: the detailed description of the CV method and different configurations and metrics used during evaluation could be moved to the methodology section.

- ll.359-360: *"The measured and interpolated results were also compared for each event in space and (r) and (rS) and the observed the interpolated spatial patterns were calculated as well"*

> This sentence makes no sense. Please reformulate!

- l.376 *"The use of KU for interpolation resulted only in a minor improvement"*

- l.379: *"In this case, OK with secondary data did not lead to an improvement"*

- ll.380-381: *"Stations located very close to each other can cause instabilities in the solution of the Kriging equations leading to high positive and negative weights"*

> Are you referring to the screening effect? Please clarify and provide a reference to a textbook to clarify what you mean by "stabilizes the solution" on l.382. KU. Would adding a nugget effect in the variograms help model the small-scale differences you see between PWS data? Please discuss!

ll.387-388: *"The poor performance  of Co-Kriging is surprising, but an appropriate selection of the co variable (for example transformed rank) may improve the results."*

> Too speculative. Please provide more details or reformulate this sentence. One explanation could be that co-kriging makes rather strong modeling assumptions (stationarity of both primary and secondary variable). It also requires the estimation and fitting of 3 (cross-)variograms, which increases uncertainty (especially in small samples). You make some other interesting comments about an extension of co-kriging toward the end of the paper. Perhaps you could include these here as well.

- ll.410-411: *"This is a typical case where all methods yield unbiased results "*

- ll.446-447: *"However, the results from this study as well as the ones from de Vos et al. (2019)"*

- ll.463-465: *"A detailed cross-validation of different filter combinations and temporal aggregations shows that the IBF is the most important step and  yields the highest improvement in interpolation quality"*

- ll.465-466: *"Furthermore, the performance of the presented method is better  at smaller temporal aggregations"*

- ll.484-485: *"Problems occur if the order of the observations is influenced by wind effects, but due to the highly skewed distribution of the precipitation amounts the problem mainly occurs for small precipitation amounts."*

> I don't understand your last argument. Please explain! The way I see it, the wind-induced bias mostly affects high rainfall intensities. Also, its effect will become more visible when quantities are aggregated over time. Wind-induced biases can represent 20-30% and are the main source of uncertainty in in-situ rainfall measurements. PWS tend to be installed in weird places and are particularly prone to this type of errors/biases.

- l.488 *"Furthermore, the near real-time availability of the data of secondary networks may help to improve the quality of flood forecasts."*

- l.490 *"on the  contrary it often increases uncertainty"*

- l.502 *"In this study, The number of primary stations  was sufficient to improve the interpolation quality"*

- ll.506-508 *"By applying a rather strict threshold of 5 C average daily temperature, many rainfall events  were rejected. It would be conceivable to include the hourly temperature data from PWS in order to estimate whether a given precipitation event corresponds to  rain or snow "*

- Figure 2: Change axis labels. For 2a, put "years" on the x-axis and "number of stations" on the y-axis.

- Figure 4: The axis labels for 4b should be "Quantile Pluvio [-]" and "Quantile Netatmo [-]"

- Equation 1: You need to specify in the equation that this only applies for Y above a certain threshold.

- Table 2: this table shows some basic statistics of the selected events and could be moved to the methods section, together with the text explaining how events were selected and how cross-validation was performed. I don't think that putting it in the results section is a good choice.

- Table 7 is interesting. But the discussion going with it is very short. You could expand this part and provide more discussion about the pros/cons of your approach compared with other faster, simpler and deterministic alternatives. I'm relieved to see that KU performs better than IDW and NN. But it's a close call and the lower performances of NN and IDW are mostly due to their higher biases compared with KU. If you would compare the methods on a fair basis, for a similar level of bias, would you still see significant differences in RMSE? Indeed, the bias in IDW can easily be reduced by performing hyperparameter optimization of the distance decay parameter or choosing a different distance metric. So there's definitively room for improvement. On the other hand, there also seems to be some room left for further optimization of the KU technique. For example, you could optimize the uncertainty parameter linked to the PWS data. In the paper, you arbitrarily use 10% but this could be tuned to the dataset as well (using LOOCV).

---

## Author Response (AR2)

*Reply to the review provided by Marc Schleiss to the revised paper*

**The use of citizen observations for better precipitation estimation and interpolation**

*submitted for publication in*
*Hydrology and Earth System Sciences*

We thank Marc Schleiss for taking his time to carefully read our paper and for his interesting discussion on the methodology. Even thought there are several points where we do not agree we appreciate his comprehensive and detailed review.

Here are our statements:

a) More details about the kriging part - The authors responded
to this comment but not of all their explanations can be found
in the revised paper. Please make sure that all important details
are in the text so that others can reproduce what you did!

Issues specifically related to the paper have been added to the text. Other topics concerning variogram scaling or discussions on local stationarity were not added, as these questions were often dealt with in other publications.

b) Comparison of kriging with simpler, faster alternatives such
as inverse distance weighted interpolation or bilinear interpolation
Partially done but results are not shown and theres only a few
short sentences in the paper about this, without any numbers
or critical discussion about the pros/cons.

It is not the aim of our paper to compare different interpolation methods in depth. In our opinion, Kriging is a standard interpolation method and if one uses an optimized code (not GIS or other custom software) time is not a problem at all. There were a great number of studies comparing IDW and Kriging for precipitation and other environmental variables, showing that Kriging outperforms IDW. Thus, we did not want to discuss these well known facts again. But as the Reviewer requests such a comparison a small table with the results is presented. Here are some recent examples comparing Kriging and IDW for precipitation:

Adhikary, Sajal Kumar Muttil, Nitin Yilmaz, Abdullah Gokhan Cokriging for enhanced spatial interpolation of rainfall in two Australian catchments *Hydrological Processes*, **31**, 21432161, 2017

Alan Mair and Ali Fares Comparison of Rainfall Interpolation Methods in a Mountainous Region of a Tropical Island JOURNAL OF HYDROLOGIC ENGINEERING, 371-382, 2011

S. Ly, C. Charles, and A. Degre, Geostatistical interpolation of daily rainfall at catchment scale:the use of several variogram models in

the Ourthe and Amblevecatchments, Belgium Hydrol. Earth Syst. Sci., 15, 22592274, 2011

Another important advantage is that the geostatistical framework allows a consistent combination of different data and variables such as Uncertainty Kriging.

1) Please clearly state the main main conclusion of your paper
   in the abstract and conclusions.  Right now, this is not
   100 % clear.  Is the conclusion that careful QC and bias-correction
   has to be performed before PWS precipitation data can be
   used?  If thats the case, then this is not really new.  Other
   studies have already shown the same and your method is just
   another way to do this.  So what exactly is your contribution?
   Please clarify!

   The novelty of our contribution is that it

   - offers a new method for finding useful PWS,

   - presents a rank based method for bias correction,

   - quantifies the improvement using PWS for interpolation.

   The conclusions of the paper were modified accordingly.

2) Your method is rather complicated.  Yet several of its components
   do not seem to significantly improve performance.  For example,
   the EBF filters and the KU do not make a big difference.
   So why did you feel the need to include them in the methods
   and results?  It just makes the paper longer and more complicated
   and forces you to introduce a lot of theory and notations
   for no obvious gain in performance.  I suggest to shorten
   the paper and only keep the essential parts of the algorithm
   in the methods section.  If you want, you can always write
   a short section or paragraph summarizing the results for
   some other options/filters that you think could be useful
   in other contexts.

   We do not think that the method is complicated, in fact it contains
   a set of simple steps.
   The two methods - the EBF filter and the KU are both useful for the
   interpolation. While EBF can under circumstances help to reduce
   the effect of false zeros, KU is more essential. KU is improving the
   interpolation in many cases.
   Concerning the KU we believe this is a very important step, which
   we consider as essential. Our arguments are as follows:

   1. PWS are even after bias correction are inferior in quality compared to the official weather service data.  Therefore it is plausible to assume an additional random error.  In fact it is very important to *downweight* these measurements due to

their uncertainty. KU is a very simple but very seldom applied method. Therefore readers should be aware of it. In our opinion KU is the correct way to handle these data, even if in our particular case it did not bring any advantage. Deterministic methods such as nearest neighbour of inverse distance do not offer a possibility to reflect data quality and are thus not our first choice.

2. When calculating normalized variograms from the weather service data only and from the bias corrected PWS only they are similar with the exception that the PWS variograms have a nugget between 10 and 25 %. This is reflected by the KU procedure.

3. We interpolated and cross validated hourly precipitation data for 7 month in 2018 and 7 in 2019 using 1,000 DWD stations and 13,000 PWS. The cross validations show that the results are much better with KU. This is of course work which was done after the submission of the paper but it confirmed our apriori assumption. It is not clear for us why our case study did not show improvements with KU, we'll have another look at it.

4. The application of KU is also related to your remark number 13.

We did a cross validation for all hourly observations from 2019. This example is now added to the paper.

3) The number of peer-reviewed studies about PWS and their use in hydrometeorology is still limited. A few of them have already been mentioned in the literature review. But overall, the introduction of the paper remains rather short. I suggest to extend this part by providing a more in-depth analysis and discussion of the state-of-the-art related to the use of citizen gauges in quantitative precipitation estimation problems, including its challenges, similarities with other fields and open questions. For example, some parts of the Discussion (i.e., the differences/similarities with radar-gauge QPE) could be moved to the introduction. Also, I encourage the authors to explicitly state which aspect(s) of the problem their study is meant to address. Whats the main contribution? Is it the method itself or is it the lessons learned and/or recommendations for a successful interpolation/merging of PWS data?

Thank you for pointing out these papers. To our knowledge, the study by de Vos et al. (2019) is the only one which uses PWS precipitation data with high temporal resolution. Other papers, like the one you mentioned don't incorporate quantitative PWS precipitation data. Therefore, we are very uneasy about these suggestions. If a paper has no influence on what we did (which is the case in the ones you suggested), why should we cite it? The last decade with high pressure on publications and citations lead to an enormous increase of the volume of introductions. Introductions are gradually becoming boring and superficial reviews. The paper we wrote is not a review paper. We intend to communicate a few new ideas which might be useful for others and not to give an overview of what else is available. Nowadays with the fast possibilities of literature search the superficial reviews are in our opinion obsolete. We still prefer short papers with clear messages like many fundamental papers written in the middle of the last century.

4) The writing and structure of the Results section need to be improved. The current strategy for assessing/validating the different components of the method is not clear to me. Right now, analyses/results are presented in seemingly random order, with rather vague qualitative statements and lots of circumstantial evidence. A better, more precise, quantitative and targeted evaluation would greatly increase the quality of the paper. For example, you could consider a step-by-step, hierarchical assessment of the different components (e.g., the IBF filter, the bias correction and the interpolation/merging), with different scores and subsections for each part.
We tried to improve the paper by restructuring the section. The usefulness of the filters and the bias correction can however be best quantified through the comparison of the cross validation results. This makes a complete step by step discussion impossible.

5) Figure A1 is crucial for understanding how the bias adjustment method works. I suggest to move this from the Appendix to the main text, together with the corresponding explanations. Actually, I dont think you need an appendix at all!
We followed this suggestion and move this figure and the corresponding explanation to the main text.

6) Table 3 does not show correlations (which should be between -1 and 1). Please correct.
It seems the reviewer did not read the table caption which is: **Percentage** of the stations with improved temporal correlation(compared to interpolation using primary stations only) for the configurations C1-C4.

7) The step-by-step description of the algorithm is a good idea. But its really hard to follow, even for somebody familiar with the geostatistical jargon. More work is needed to streamline this and make it clear. A flowchart of the whole method would help, with different symbols for filters, adjustments and interpolations! Also, you could shorten the text by grouping some of the smaller steps together into larger modules

or tasks.  The details of each task can be given in the different
subsections of the methodology.
A flow chart summarizing the steps of the procedure starting with
the indicator filter and ending with the interpolation procedure was
added.

8) The crucial assumption behind your method is that for high
   precipitation intensities, the ranks of the secondary stations
   are correct.  Some superficial analyses in Section 4.1 suggest
   that this assumption is probably not too bad.  But since
   this is such a critical hypothesis, it should be assessed
   in much more detail.  Please extend Section 4.1 and perform
   more tests designed to assess how good this ordering assumption
   really is.  For example, your could compute rank correlation
   coefficients for different thresholds, stations and lengths
   of time series.  Or you could look at fluctuations over time
   or as a function of distance.  To better understand the limitations
   of your method, it could also be good to show a few cases
   for which the assumption does not hold.

   The rank correlations for close stations were calculated for pairs of
   primary and secondary stations closer than 2500m to each other,
   separately for stations which were removed by the indicator filter,
   and those which were not removed. Their histograms are presented
   in the subsection discussing the filter, as the results both support
   the hypothesis and the usefulness of the indicator filter.

9) I have some issues with the terminology chosen by the authors,
   especially regarding the EBF (Eventbased filter).  I think
   this is a poor choice of words.  In reality, the EBF filter
   is a spatial filter for one particular aggregation time period
   (and not an event).  More generally, I dont think that it
   is a good idea to use the word event to refer to a particular
   aggregation time periods.  This is not standard practice
   and might be confusing to many readers.  Please modify accordingly.
   Both filters are mainly spatial filters, using observations of close sta-
   tions. The difference is that the first filter is using the whole time
   series of a particular PWS, while the second is used for the inves-
   tigation of a particular time step (not aggregation). Therefore, we
   used the word event to make this distinction clear.

10) Regarding the bias correction scheme:  If I understood the
    approach correctly, the idea is to use the percentile of
    the PWS observations (secondary network) to estimate the
    equivalent precipitation estimates of the professional gauges
    (primary network) and then spatially interpolate this value
    to the location of the PWS using kriging.  On top of the
    large uncertainty that comes with estimating a percentile
    from a short PWS series, one problem with this approach is

that it uses the ordering assumption multiple times (i.e.,
once for each pair or PWS and professional gauge).  This
greatly increases the chances of errors during bias correction
due to imperfect modeling assumptions.  Also, the final spatial
interpolation may re-introduce bias due to smoothing and/or
modeling choices.  So my question is:  why dont you just
pool the professional rain gauge data together into a single
distribution and directly adjust the PWS observations using
quantile-quantile mapping on the pooled data?  In this way,
you would use the ordering assumption only once and you would
not have to interpolate at all, which is likely to be faster
and more robust.  By the way, you can pool data even if the
time series of the professional gauges have different lengths.
Please explain why you think the current approach is better!

The reviewer seems to have partly misunderstood the idea.  We
do not assume that the order at the PWS and the closest DWD
station is the same. We assume that if the precipitation measured
by a PWS at time $t_1$ is larger than the precipitation measured at
the same location at time $t_2$ then the real (unknown) precipitation
at the location of the PWS at time $t_1$ was also larger than at time $t_2$.
This does not involve the professional gauges at all. In equation (1)
- where $Z$ is the precipitation which was not measured at location
$y_i$.  Here there is no assumption on the primary network.  The
primary network is used to estimate the distribution function of
precipitation at the PWS location. The sample size is not a major
problem.  Using 7 month (snow free) hourly data we have 5136
observations, which is a much bigger sample then often used in
hydrological applications. For larger aggregations (for example 24
hours) the bias correction should be done on an hourly basis and
aggregated afterwards.

Pooling all data is not a good alternative as some of the stations
may have a positive, while others a negative bias. (For example
due to manual calibration of the device.)  If one pools all data
then these partly visible differences cannot be considered. We do
have PWS with systematic bias which is clearly visible if one com-
pares monthly or seasonal sums with the interpolated sums of the
primary stations. The bias exceeds often 20 %, and both over and
underestimations occur. The method suggested would preserve this
bias.

As this was a suggestion of the reviewer, and was not directly con-
sidered we do not think that this idea has to be discussed in the
paper.

11) A substantial part of Section 5 (Discussion) from lines 434-455
is not a discussion but just a summary of the method and
therefore should be moved to the conclusions.  The last part
of the discussion (ll.467-475) about the similarities/differences

of PWS with radar measurements.  This is out of scope here
because not part of the analyses.  I suggest to shorten this
and/or move it to the introduction.  Please use the discussion
section to analyze pros/cons, mention alternatives or new
ideas for follow-up studies.
We've rewritten the discussion accordingly. For the sake of read-
ability, we've not use track changes for this section.

12) Conclusions, ll.501-503:  Wind has a major effect on precipitation
measurements, leading to a systematic undercatch.  This may
influence the order of data, but the effect is the same for
the primary and secondary network.
I do not agree with this statement.  Literature shows that
wind effects tend to be very local.  Sometimes, both gauges
will be affected by the same bias.  But often, its likely
that the PWS and professional gauges will have different
biases.  More importantly, wind-induced biases will fluctuate
over time and space, which affects the rank statistics and
the performance of the IBF and bias correction schemes.  Theres
not much that you can do about this.  But at least, you should
properly acknowledge the problem and discuss its possible
consequences in the text.  I suggest to do this in Section
5 (Discussion) rather than the conclusions.
Please note that the bias correction does not use simultaneous ob-
servations of the primary and PWS network for the bias correction.
Therefore whether they have the same or a different wind influence
is not of great importance. Problems occur if the order of the obser-
vations is influenced by wind effects, but due to the highly skewed
distribution of the precipitation amounts the problem mainly oc-
curs for small precipitation amounts. A related statement is moved
to the discussion.

13) On a personal note:  PWS stations tend to cluster in/around
urban areas.  Spatial interpolation methods such as kriging
do not always perform optimally on highly clustered data.
For example, it is well known that clustering can lead to
screening effects and highly negative kriging weights.  This
does not necessarily lead to wrong estimates but decreases
robustness and accuracy.  I am aware that this goes beyond
the scope of this study.  Still, I invite the authors to
briefly mention this issue in the Discussion section and
to point to possible ways to overcome it in future work.
This is particularly relevant for small-scale estimates of
heavy precipitation.
As mentioned in the reply to comment 2) the consideration of ob-
servation uncertainty in the Kriging procedure for the PWS solves
the problem, and thus UK is important for possible applications.
A short statement concerning this problem is added to the paper.

Here is an example showing this effect: we've prepared (extracted) a little example for you showing that KU can help to overcome the problem of unstable (negative) weights in a reasonable way:
For a given (real) configuration of 2 primary stations and 7 PWS (some of them clustered) the Ordinary Kriging equations are:

$$
\begin{bmatrix}
1.000 & 0.023 & 0.538 & 0.559 & 0.666 & 0.796 & 0.637 & 0.910 & 0.353 & 1.000 \\
0.023 & 1.000 & 0.134 & 0.127 & 0.080 & 0.047 & 0.054 & 0.039 & 0.115 & 1.000 \\
0.538 & 0.134 & 1.000 & 0.698 & 0.949 & 0.821 & 0.844 & 0.747 & 0.392 & 1.000 \\
0.559 & 0.127 & 0.698 & 1.000 & 0.626 & 0.556 & 0.470 & 0.601 & 0.844 & 1.000 \\
0.666 & 0.080 & 0.949 & 0.626 & 1.000 & 0.951 & 0.956 & 0.883 & 0.330 & 1.000 \\
0.796 & 0.047 & 0.821 & 0.556 & 0.951 & 1.000 & 0.958 & 0.969 & 0.289 & 1.000 \\
0.637 & 0.054 & 0.844 & 0.470 & 0.956 & 0.958 & 1.000 & 0.865 & 0.226 & 1.000 \\
0.910 & 0.039 & 0.747 & 0.601 & 0.883 & 0.969 & 0.865 & 1.000 & 0.336 & 1.000 \\
0.353 & 0.115 & 0.392 & 0.844 & 0.330 & 0.289 & 0.226 & 0.336 & 1.000 & 1.000 \\
1.000 & 1.000 & 1.000 & 1.000 & 1.000 & 1.000 & 1.000 & 1.000 & 1.000 & 0.000
\end{bmatrix}
x =
\begin{bmatrix}
0.421 \\ 0.225 \\ 0.906 \\ 0.817 \\ 0.763 \\ 0.610 \\ 0.606 \\ 0.571 \\ 0.547 \\ 1.000
\end{bmatrix}
$$

leads to the solution:

$$
\begin{bmatrix}
0.034 & 0.038 & 2.194 & 0.406 & -2.513 & 2.072 & -0.269 & -0.874 & -0.089 & 0.002
\end{bmatrix}
$$

Due to the high positive and negative weights makes the estimator very unstable.
Using the uncertainty kriging approach assuming a 10 % variance increase due to the uncertainty of the PWS leads to the equation system

$$
\begin{bmatrix}
1.000 & 0.023 & 0.538 & 0.559 & 0.666 & 0.796 & 0.637 & 0.910 & 0.353 & 1.000 \\
0.023 & 1.000 & 0.134 & 0.127 & 0.080 & 0.047 & 0.054 & 0.039 & 0.115 & 1.000 \\
0.538 & 0.134 & \mathbf{1.100} & 0.698 & 0.949 & 0.821 & 0.844 & 0.747 & 0.392 & 1.000 \\
0.559 & 0.127 & 0.698 & \mathbf{1.100} & 0.626 & 0.556 & 0.470 & 0.601 & 0.844 & 1.000 \\
0.666 & 0.080 & 0.949 & 0.626 & \mathbf{1.100} & 0.951 & 0.956 & 0.883 & 0.330 & 1.000 \\
0.796 & 0.047 & 0.821 & 0.556 & 0.951 & \mathbf{1.100} & 0.958 & 0.969 & 0.289 & 1.000 \\
0.637 & 0.054 & 0.844 & 0.470 & 0.956 & 0.958 & \mathbf{1.100} & 0.865 & 0.226 & 1.000 \\
0.910 & 0.039 & 0.747 & 0.601 & 0.883 & 0.969 & 0.865 & \mathbf{1.100} & 0.336 & 1.000 \\
0.353 & 0.115 & 0.392 & 0.844 & 0.330 & 0.289 & 0.226 & 0.336 & \mathbf{1.100} & 1.000 \\
1.000 & 1.000 & 1.000 & 1.000 & 1.000 & 1.000 & 1.000 & 1.000 & 1.000 & 0.000
\end{bmatrix}
x =
\begin{bmatrix}
0.421 \\ 0.225 \\ 0.906 \\ 0.817 \\ 0.763 \\ 0.610 \\ 0.606 \\ 0.571 \\ 0.547 \\ 1.000
\end{bmatrix}
$$

leads to the *domesticated* solution:

$$
\begin{bmatrix}
-0.083 & 0.103 & 0.631 & 0.393 & 0.128 & -0.056 & -0.077 & -0.042 & 0.003 & -0.012
\end{bmatrix}
$$

**Minor Comments**

We've corrected the typos, reformulated the sentences and implemented most of the remarks. Here's our response to the remaining comments:

- Introduction, ll.24-26, This is potentially very useful to complement systematic weather observations of national weather services, especially with respect to precipitation, which is highly variable in space and time. Please add a few references at the end of this sentence to support your statement.

The fact that precipitation is variable in space and time is common knowledge and does not need to be referenced from our point of view.

- Section 2, l.69, The gauges used in this network are typically weighing gauges.Do you mean predominantly?  In addition, please specify the type of weighing gauges (e.g., the model, brand or serial number).

Yes, we mean predominantly. The fact that most gauges are weighing gauges should be sufficient information for the readers. Anyone who how is interested in more technical details about the rain gauges can contact the German Weather Service.

Figure 3:  Please use different symbols for N07, N10 and N11 to better distinguish the points.

We do not wish to change this as different symbols as they would not make the points more distinguishable. The message that this figure conveys is that the extreme scatter less than the lower values.

- Figure 1:  Please add a scale!  Same comment for figures 6, 8, 9,10

Done!

- Figure 4:  Please specify the 3 primary and 4 secondary stations in the caption and how far away they are from each other.

What exactly do you mean by specify? We've added information about the distances between these stations in the figure caption.

- On l.119, you mention that the random variable Y is not stationary. Yet, on ll.144-145 and Equation 2, you refer to its cumulative distribution function F, without any dependence on time.  Please clarify this apparent contradiction.

The rank assumption (1) means that even if $Y$ is not stationary its indicator is. There is no contradiction here. The distribution function $F$ corresponds to the spatially stationary variable $Z$. The distribution function $G$ is defined for each secondary PWS over time. For this we do not need any spatial stationarity. The word spatially was added to the sentence to make this issue clear.

Equation 4, whats your definition of nearly at the same separation? Please specify!

As the spacing of the primary network is different and in order to take the natural variability of the indicator correlations in space we use a window around the selected distance - similarly as for variogram calculations. Some clarification was added, both in text and equation.

- ll.168-172, Under the assumption that the temporal order of precipitation at secondary is correct (eq.1), one could have used rank correlations instead of the indicator correlations. The indicator approach is preferred however, as the sensitivity of the devices of the primary and secondary networks is different

and this would influence the order of the small values strongly.
Furthermore, random measurement errors would also influence the
order of low values.  In order to have a sufficient sample size
and to have robust results, high $\alpha$ values and low temporal aggregations
$\Delta t$ tare preferred.

Or you could just say that the ordering between the primary and
secondary networks needs to be the same for values above a certain
threshold.

No, the temporal order (even for intense precipitation) at the primary
and the secondary stations can be different simply due to the spatial
variability of precipitation. However for intense precipitation the extent
of the rainfall field is usually large enough so that nearby stations both
have high ranks.

- Section 5, ll.452-455 The use of secondary stations after filtering
and data transformation improves the results of interpolation
for other possible interpolation methods, such as nearest neighbour
or inverse distance weighting.  However, in this study these
methods yield worse results than OK (results not shown here).
Not clear.  Please provide more details.  For example, you could
give the average reduction in terms of RMSE or increase in correlation
for each interpolation method.

A table concerning the improvement for other interpolation methods was
added to the paper, even thought the focus was not on the comparison
of the interpolation methods.

[revised manuscript text omitted]

---

## Author Response (AR3)

Reply to the review provided by Marc Schleiss to the third revision of the paper

**The use of citizen observations for better precipitation estimation and interpolation**

submitted for publication in Hydrology and Earth System Sciences

We thank Marc Schleiss for taking his time to carefully reread our paper and for his remarks. We appreciate his thorough review, but there are also some points where we do not agree.

Here are our statements:

This is the third time that I review this paper. The paper has improved. However, there are still a lot of typos and unclear sentences and the writing could be improved further. Most of the major issues I raised during the previous rounds were (partially) addressed. The only major points of criticism that I have left are:

- the justification of the assumptions in Section 4.1, which could be more quantitative and exhaustive

In the revised version we already added a new Figure 7 - showing the rank correlations of the closest PWD and primary station pairs. This figure is in Section 4.2 as it both supports the hypothesis of section 4.1 and the indicator based filter. One could add further justifications, but we did not want to extend the paper which is already quite long.

- the conclusion section, which is too short and does not include all major findings.

The conclusion section was changed.....

- an assumption on the rank stability of the PWS stations was introduced.

- A Kriging interpolation with uncertain data was used (KU). This method allows the downweighting of the PWS stations, and leads to an improvement of the interpolation quality.

- the structure of the paper. In particular, the event selection procedure and cross-validation strategies which should not be in the results part but introduced earlier in the text, in the methods section.

The novelty of this paper is described in the methodology chapter. The cross validation procedure is not new and used only for the evaluation of the results. That's why we put it in the results section. Furthermore, we consider this a question of personal taste. Restructuring the paper does not change its scientific content, therefore we would like to leave it as it is.

"Although this is only one example with a relatively short time period it does support our assumption that the quantiles between primary and secondary stations are similar for higher precipitation intensities. However, one secondary device (N10) delivered data which deviates substantially from the other measurements. This was caused by an interrupted connection between the rain sensor and the base station. In this case, the total sum of precipitation over a longer time period was transferred at once (i.e. in one single measurement interval) when the connection was established again. This leads to an extreme outlier which falsifies the results."

Not sure to understand your argument here. According to your assumption, the points in Figure 4b should align with each other (though not necessarily along y=x). Still, there seems to be substantial residual scatter and uncertainty due to quantization effects (especially for Netatmo). Please provide some quantitative metrics to judge the degree of linear relationship and highlight which data point in 4b corresponds to the "extreme outlier". In addition, it would be worth commenting on the discretization effects you see in the Netatmo stations.

In fact the points on figure correspond to the ranks of the observations and they should be on the line y=x. The residual scatter on Figure 4b is likley due to data transmission issues of Netatmo station N10. Apparently, this station in particular often failed to transmit data during rain events and hence transmitted data later on when it was dry. Therefore, the extreme outlier is not displayed in the plot since it occurred at at time when the reference recorded 0mm rainfall (i.e. it would be located at the 0 quantile of the reference an the highest quantile of the PWS station). In this figure we wanted to show that PWS can behave erratically and such stations need to be detected and removed (even if they are placed correctly). The indicator filter recognizes this error and suggests the removal of this station. We reformulated this section to make this more clear

11.73-74: corrected

11.103-104: corrected

11.105-107: corrected

11.119: corrected

ll.149-150: In order to have a sufficient sample size and to have robust results, high  $\alpha$  values and low temporal aggregations  $\Delta t$  are preferred.

Can you be more specific? What are sufficiently large values for alpha and delta t?

In order to have a sufficient sample size and to have robust results  $\alpha$  and

 $\Delta t$  have to be selected carefully. The lower the  $\Delta t$  value is the more observations are available, but the number of zeros also increases. Thus the corresponding  $\alpha$  has increase. The selection of  $\Delta t = 1$  hour and  $\alpha = 0.99$  is in our case a good choice.

11.163-164: "Another possibility is to interpolate the quantiles corresponding to selected non percentiles or interpolating percentiles for selected precipitation amounts." Not clear. Please reformulate. Corrected: Another possibility is to interpolate the quantiles corresponding to selected percentiles or interpolating percentiles for selected precipitation amounts.

11.272-273: corrected

1.276: "Figure 4 shows that for high percentiles their occurrence is the same for the primary and the secondary devices." This sentence is not clear. Please reformulate.

11.272–273: done, section 4.1. is partially rewritten

11.289-291 "While the distributions differ, the probability of no precipitation p0 (defined as precipitation < 0.1 mm) ranges from 0.90 to 0.91 and is thus very similar for both types of stations indicating that the occurrence of precipitation can be well detected by the secondary network." Actually, in Table 1, the percentages p0 (at 1h resolution) are 0.84 for N07 and N10, which is 7% lower than for N11 (0.91) and 8% lower than for the Pluvio (0.92). Please explain!

This statement is referring to the analysis in Reutlingen and not Table 1. We see that is leads to a misunderstanding hence we added a subplot showing the p0 to Fig 5 and changed the caption accordingly. The fact that the p0 at our weather station are so not well captured could be related the the short time series at this location. We've also corrected a typo in Table 1: p0 for N10 at 1h is 0.94

1.302 "In our case, 862 secondary stations remained after the application of the IBF." In addition to the number, please specify the percentage of stations that were removed.

We've added some numbers in the manuscript

**11.316-317: corrected**

ll.332-334 "The cross validation was carried out for a set of different temporal aggregations  $\Delta t$  and a set of selected events. Only times with intense precipitation were selected, as for low-intensity cases the interpolation based on the primary network is sufficiently accurate" Actually, you did not show any evidence that the interpolation for lower intensities is accurate. Please provide some numbers or reformulate this sentence. We are convinced that this is a valid argument but since we did not check or quantify this we've deleted the last part of this sentence.

tt ll.338-348: the detailed description of the CV method and different configurations and metrics used during evaluation could be moved to the methodology section.

cf. answer above. This is again matter of taste and personal style. We would like to leave it as it is.

11.359-360: The measured and interpolated results were also compared for each event in space and (r) and (rS) and the observed the interpolated spatial patterns were calculated as well This sentence makes no sense. Please reformulate!

You are right, we changed it as follows: The measured and interpolated results were also compared for each event in space and the correlations between the observed and the interpolated spatial patterns were calculated as well.

**1.376 corrected**

1.379 corrected

11.380-381: Stations located very close to each other can cause instabilities in the solution of the Kriging equations leading to high positive and negative weights Are you referring to the screening effect? Please clarify and provide a reference to a textbook to clarify what you mean by stabilizes the solution on 1.382. KU. Would adding a nugget effect in the variograms help model the small-scale differences you see between PWS data? Please discuss!

This sentence was added in response to the reviewers previous review. An example illustrating the effect was enclosed. The problem is that if two stations are very close to each other and the nugget is low (or zero) then the corresponding columns of the Kriging matrix are nearly identical - which leads to high condition numbers of the corresponding matrix and thus instabilities in the solution of the linear equation system. If a measurement error term (or nugget) is applied to the diagonal, the conditioning number decreases. This is not related to the screening effect it is linear algebra. The conditioning numbers of the Kriging equations were discussed for example in:

Davis, G.J., Morris, M.D. Six factors which affect the condition number of matrices associated with kriging. Math Geol 29, 669683 (1997). https://doi.org/10.1007/BF02769650

We changed the text to:

11.387-388: The poor performance of Co-Kriging is surprising,

but an appropriate selection of the co variable (for example transformed rank) may improve the results.

Too speculative. Please provide more details or reformulate this sentence. One explanation could be that co-kriging makes rather strong modelling assumptions (stationarity of both primary and secondary variable). It also requires the estimation and fitting of 3 (cross-)variograms, which increases uncertainty (especially in small samples). You make some other interesting comments about an extension of co-kriging toward the end of the paper. Perhaps you could include these here as well.

The Co-Kriging and the other interpolation methods were performed and added to the revised version of the paper. We did not want to include a long discussion here, but we added the following text in order to explain the problem.

The poor performance of the Co-Kriging is surprising. For this study we used the PWS observations a co-variable. The linear relationship which is supposed to exisit between the investigated variable (precipitation) and the secondary variable (precipitation measured at PWS) for the application of Co-Kriging may not be appropriate for this pair of variables. Considering the ranks of the PWS observations or other transformed values as co-variables may improve Co-Kriging results, but this is not the primary topic of this paper.

11.410-411 corrected

11.446-447 corrected

11.463-465 corrected

11.465-466 corrected

- 11.484-485: Problems occur if the order of the observations is influenced by wind effects, but due to the highly skewed distribution of the precipitation amounts the problem mainly occurs for small precipitation amounts.

I dont understand your last argument. Please explain! The way I see it, the wind-induced bias mostly affects high rainfall intensities. Also, its effect will become more visible when quantities are aggregated over time. Wind-induced biases can represent 20-30% and are the main source of uncertainty in in-situ rainfall measurements. PWS tend to be installed in weird places and are particularly prone to this type of errors/biases.

**we changed the text to:**

The suggested methodology uses ranks and not the measured precipitation values of the PWS. Thus, the problem related to wind only affects the results if it changes the order of the precipitation measured at the same location. This order however is relatively stable for high precipitation values, as due to the skewness of the distribution the difference between the measured values is high.

1.488 corrected

1.490 corrected

1.502 corrected

11.506-508 corrected

Figure 2: Change axis labels. For 2a, put years on the x-axis and number of stations on the y-axis.

done

Figure 4: The axis labels for 4b should be Quantile Pluvio [-] and Quantile Netatmo [-]

**done**

Equation 1: You need to specify in the equation that this only applies for Y above a certain threshold.

This is the hypothesis, the higher values are emphasized as random errors have less influence on their true ranks

Table 2: this table shows some basic statistics of the selected events and could be moved to the methods section, together with the text explaining how events were selected and how cross-validation was performed. I dont think that putting it in the results section is a good choice.

See previous comments, we would like to leave as it is.

Table 7 is interesting. But the discussion going with it is very short. You could expand this part and provide more discussion about the pros/cons of your approach compared with other faster, simpler and deterministic alternatives. Im relieved to see that KU performs better than IDW and NN. But its a close call and the lower performances of NN and IDW are mostly due to their higher biases compared with KU. If you would compare the methods on a fair basis, for a similar level of bias, would you still see significant differences in RMSE? Indeed, the bias in IDW can easily be reduced by performing hyperparameter optimization of the distance decay parameter or choosing a different distance metric. So theres definitively room for improvement. On the other hand, there also seems to be some room left for further optimization of the KU technique. For example, you could optimize the uncertainty parameter linked to the PWS data. In the paper, you arbitrarily use 10% but this could be tuned to the dataset as well (using LOOCV).

We do not consider Kriging as a very complicated interpolation procedure. IDW and NN are easier to use, but in the early days of geostatistics several comparisons of Kriging with IDW and NN showed a superiority of Kriging. We do not want to repeat this discussion in this paper. It is however interesting to observe that after the selection of the appropriate PWS and the bias correction of their values, all methods improve. Removing the bias of the NN and the IDW does not change the order of the quality of estimators. The 10 % error was used after comparing variograms calculated from the primary and the PWS stations separately. The detailed variogram analysis was not included in this paper in order to keep it at an accaptable length. A LOOCV based estimation would certainly be a good alternative. Discussion of the Co-Kriging results was added as described above.

[revised manuscript text omitted]